# On the Stochastic (Variance-Reduced) Proximal Gradient Method for Regularized Expected Reward Optimization

**Ling Liang**                                                      *liang.ling@u.nus.edu*
*Department of Mathematics*
*University of Maryland at College Park*

**Haizhao Yang**                                                    *hzyang@umd.edu*
*Department of Mathematics and Department of Computer Science*
*University of Maryland at College Park*

**Reviewed on OpenReview:** *https://openreview.net/forum?id=Ve4Puj2LVT*

## Abstract

We consider a regularized expected reward optimization problem in the non-oblivious setting that covers many existing problems in reinforcement learning (RL). In order to solve such an optimization problem, we apply and analyze the classical stochastic proximal gradient method. In particular, the method has shown to admit an $O(\epsilon^{-4})$ sample complexity to an $\epsilon$-stationary point, under standard conditions. Since the variance of the classical stochastic gradient estimator is typically large, which slows down the convergence, we also apply an efficient stochastic variance-reduce proximal gradient method with an importance sampling based ProbAbilistic Gradient Estimator (PAGE). Our analysis shows that the sample complexity can be improved from $O(\epsilon^{-4})$ to $O(\epsilon^{-3})$ under additional conditions. Our results on the stochastic (variance-reduced) proximal gradient method match the sample complexity of their most competitive counterparts for discounted Markov decision processes under similar settings. To the best of our knowledge, the proposed methods represent a novel approach in addressing the general regularized reward optimization problem.

## 1 Introduction

Reinforcement learning (RL) (Sutton & Barto, 2018) has recently become a highly active research area of machine learning that learns to make sequential decisions via interacting with the environment. In recent years, RL has achieved tremendous success so far in many applications such as control, job scheduling, online advertising, and game-playing (Zhang & Dietterich, 1995; Pednault et al., 2002; Mnih et al., 2013), to mention a few. One of the central tasks of RL is to solve a certain (expected) reward optimization problem for decision-making. Following the research theme, we consider the following problem of maximizing the regularized expected reward:

$$\max_{\theta \in \mathbb{R}^n} \; \mathcal{F}(\theta) := \mathbb{E}_{x \sim \pi_\theta}\left[\mathcal{R}_\theta(x)\right] - \mathcal{G}(\theta), \tag{1}$$

where $\mathcal{G} : \mathbb{R}^n \to \mathbb{R} \cup \{+\infty\}$ is a closed proper convex (possibly nonsmooth) function, $x \in \mathbb{R}^d$, $\mathcal{R}_\theta : \mathbb{R}^d \to \mathbb{R}$ is the reward function depending on the parameter $\theta$, and $\pi_\theta$ denotes the probability distribution over a given subset $\mathcal{S} \subseteq \mathbb{R}^d$ parameterized by $\theta \in \mathbb{R}^n$. By adapting the convention in RL, we call $\pi_\theta$ a policy parameterized by $\theta$. Moreover, for the rest of this paper, we denote $\mathcal{J}(\theta) := \mathbb{E}_{x \sim \pi_\theta}\left[\mathcal{R}_\theta(x)\right]$ as the expected reward function in the *non-oblivious* setting. The learning objective is to learn a decision rule via finding the policy parameter $\theta$ that maximizes the regularized expected reward. To the best of our knowledge, the study on the general model (1) has been limited in the literature. Hence, developing and analyzing algorithmic frameworks for solving the problem is of great interest.

There are large body of works in supervised learning focusing on the *oblivious* setting (Zhang, 2004; Hastie et al., 2009; Shapiro et al., 2021), i.e., $\mathcal{J}(\theta) := \mathbb{E}_{x \sim \pi}\left[\mathcal{R}_\theta(x)\right]$, where $x$ is sampled from an invariant distri-

bution $\pi$. Clearly, problem (1) can be viewed as a generalization of those machine learning problems with oblivious objective functions. In the literature, an RL problem is often formulated as a discrete-time and discounted Markov decision processes (MDPs) (Sutton & Barto, 2018) which aims to learn an optimal policy via optimizing the (discounted) cumulative sum of rewards. We can also see that the learning objective of an MDP can be covered by the problem (1) with the property that the function $\mathcal{R}(x)$ does not depend on $\theta$ (see Example 3.3). Recently, the application of RL for solving combinatorial optimization (CO) problems which are typically NP-hard has attracted much attention. These CO problems may include the traveling salesman problem and related problems (Bello et al., 2016; Mazyavkina et al., 2021), the reward optimization problem arising from the finite expression method (Liang & Yang, 2022; Song et al., 2023), and the general binary optimization problem (Chen et al., 2023), to name just a few. The common key component of the aforementioned applications is the reward optimization, which could also be formulated as problem (1). There also exist problems with general reward functions that are outside the scope of cumulative sum of rewards of trajectories that are used in MDPs. An interesting example is the MDP with general utilities; see, e.g., (Zhang et al., 2020a; Kumar et al., 2022; Barakat et al., 2023) and references therein.

Adding a regularizer to the objective function is a commonly used technique to impose desirable structures to the solution and/or to greatly enhance the expression power and applicability of RL (Lan, 2023; Zhan et al., 2023). When one considers the direct/simplex parameterization (Agarwal et al., 2021) of $\pi_\theta$, a regularization function using the indicator function for the standard probability simplex is needed. Moreover, by using other indicator functions for general convex sets, one is able to impose some additional constraints on the parameter $\theta$. For the softmax parameterization, one may also enforce a bounded constraint to $\theta$ to prevent it taking values that are too large. This can avoid potential numerical issues, including the overflow error on a floating point system. On the other hand, there are incomplete parametric policy classes, such as the log-linear and neural policy classes, that are often formulated as $\{\pi_\theta | \theta \in \Theta\}$, where $\Theta$ is a closed convex set (Agarwal et al., 2021). In this case, the indicator function is still necessary and useful. Some recent works (see, e.g., (Ahmed et al., 2019; Agarwal et al., 2020; Mei et al., 2020; Cen et al., 2022)) have investigated the impact of the entropy regularization for MDPs. Systematic studies on general convex regularization for MDPs have been limited until the recent works (Pham et al., 2020; Lan, 2023; Zhan et al., 2023). Finally, problem (1) takes the same form as the stochastic optimization problem with decision-dependent distributions (see e.g., (Drusvyatskiy & Xiao, 2023) and references therein), leading to numerous real-world applications such as performative prediction (Mendler-Dünner et al., 2020; Perdomo et al., 2020), concept drift (Gama et al., 2014), strategic classification (Tsirtsis et al., 2024; Milli et al., 2019), and casual inference (Yao et al., 2021). Consequently, we can see that problem (1) is in fact quite general and has promising modeling power, as it covers many existing problems in the literature.

The purpose of this paper is to leverage existing tools and results in MDPs and nonconvex optimization for solving the general regularized expected reward optimization problem (1) with general policy parameterization, which, to the best of our knowledge, has not been formally considered in the RL literature. It is well known that the policy gradient method (Williams, 1992; Sutton et al., 1999; Baxter & Bartlett, 2001), which lies in the heart of RL, is one of the most competitive and efficient algorithms due to its simplicity and versatility. Moreover, the policy gradient method is readily implemented and can be paired with other effective techniques. In this paper, we observe that the stochastic proximal gradient method, which shares the same spirit of the policy gradient method, can be applied directly for solving the targeted problem (1) with convergence guarantees to a stationary point. Since the classical stochastic gradient estimator typically introduces a large variance, there is also a need to consider designing advanced stochastic gradient estimators with smaller variances. To this end, we shall also look into a certain stochastic variance-reduced proximal gradient method and analyze its convergence properties. In particular, the contributions of this paper are summarized as follows.

- We consider a novel and general regularized reward optimization model (1) that covers many existing important models in the machine learning and optimization literature. Thus, problem (1) admits a promising modeling power which encourages potential applications.

- In order to solve our targeted problem, we consider applying the classical stochastic proximal gradient method and analyze its convergence properties. We first demonstrate that the gradient of $\mathcal{J}(\cdot)$ is Lipschitz continuous under standard conditions with respect to the reward function $\mathcal{R}_\theta(\cdot)$ and

the parameterized policy $\pi_\theta(\cdot)$. Using the L-smoothness of $\mathcal{J}(\cdot)$, we then show that the classical stochastic proximal gradient method with a constant step-size (depending only on the Lipschitz constant for $\nabla_\theta \mathcal{J}(\cdot)$) for solving problem (1) outputs an $\epsilon$-stationary point (see Definition 3.4) within $T := O(\epsilon^{-2})$ iterations, and the sample size for each iteration is $O(\epsilon^{-2})$, where $\epsilon > 0$ is a given tolerance. Thus, the total sample complexity becomes $O(\epsilon^{-4})$, which matches the current state-of-the-art sample complexity of the classical stochastic policy gradient for MDPs; see e.g., (Williams, 1992; Baxter & Bartlett, 2001; Zhang et al., 2020b; Xiong et al., 2021; Yuan et al., 2022).

- Moreover, in order to further reduce the variance of the stochastic gradient estimator, we utilize an importance sampling based probabilistic gradient estimator which leads to an efficient single-looped variance reduced method. The application of this probabilistic gradient estimator is motivated by the recent progress in developing efficient stochastic variance-reduced gradient methods for solving stochastic optimization (Li et al., 2021b) and (unregularized) MDPs (Gargiani et al., 2022). We show that, under additional technical conditions, the total sample complexity is improved from $O(\epsilon^{-4})$ to $O(\epsilon^{-3})$. This result again matches the results of some existing competitive variance-reduced methods for MDPs (Papini et al., 2018; Xu et al., 2019; Pham et al., 2020; Huang et al., 2021; Yang et al., 2022; Gargiani et al., 2022). Moreover, to the best of our knowledge, the application of the above probabilistic gradient estimator is new for solving the regularized expected reward optimization (1).

The rest of this paper is organized as follows. We first summarize some relative works in Section 2. Next, in Section 3, we present some background information that are needed for the exposition of this paper. Then, in Section 4, we describe the classical stochastic proximal gradient method for solving (1) and present the convergence properties of this method under standard technical conditions. Section 5 is dedicated to describing and analyzing the stochastic variance-reduced proximal gradient method with an importance sampling based probabilistic gradient estimator. Finally, we make some concluding remarks, and list certain limitations and future research directions of this paper in Section 6.

## 2 Related Work

**The policy gradient method**. One of the most influential algorithms for solving RL problems is the policy gradient method, built upon the foundations established in (Williams, 1992; Sutton et al., 1999; Baxter & Bartlett, 2001). Motivated by the empirical success of the policy gradient method and its variants, analyzing the convergence properties for these methods has long been one of the most active research topics in RL. Since the objective function $\mathcal{J}(\theta)$ is generally nonconcave, early works (Sutton et al., 1999; Pirotta et al., 2015) focused on the asymptotic convergence properties to a stationary point. By utilizing the special structure in (entropy regularized) MDPs, recent works (Liu et al., 2019; Mei et al., 2020; Agarwal et al., 2021; Li et al., 2021a; Xiao, 2022; Cen et al., 2022; Lan, 2023; Fatkhullin et al., 2023) provided some exciting results on the global convergence. Meanwhile, since the exact gradient of the objective function can hardly be computed, sampling-based approximated/stochastic gradients have gained much attention. Therefore, many works investigated the convergence properties, including the iteration and sample complexities, for these algorithms with inexact gradients; see e.g., (Zhang et al., 2020b; Liu et al., 2020; Zhang et al., 2021b; Xiong et al., 2021; Yuan et al., 2022; Lan, 2023) and references therein.

**Variance reduction**. While the classical stochastic gradient estimator is straightforward and simple to implement, one of its most critical issues is that the variance of the inexact gradient estimator can be large, which generally slows down the convergence of the algorithm. To alleviate this issue, an attractive approach is to pair the sample-based policy gradient methods with certain variance-reduced techniques. Variance-reduced methods were originally developed for solving (oblivious) stochastic optimization problems (Johnson & Zhang, 2013; Nguyen et al., 2017; Fang et al., 2018; Li et al., 2021b) typically arising from supervised learning tasks. Motivated by the superior theoretical properties and practical performance of the stochastic variance-reduced gradient methods, similar algorithmic frameworks have recently been applied for solving MDPs (Papini et al., 2018; Xu et al., 2019; Yuan et al., 2020; Pham et al., 2020; Huang et al., 2021; Yang et al., 2022; Gargiani et al., 2022).

**Stochastic optimization with decision-dependent distributions**. Stochastic optimization is the core of modern machine learning applications, whose main objective is to learn a decision rule from a limited data sample that is assumed to generalize well to the entire population (Drusvyatskiy & Xiao, 2023). In the classical supervised learning framework (Zhang, 2004; Hastie et al., 2009; Shapiro et al., 2021), the underlying data distribution is assumed to be static, which turns out to be a crucial assumption when analyzing the convergence properties of the common stochastic optimization algorithms. On the other hand, there are problems where the distribution changes over the course of iterations of a specific algorithm, and these are closely related to the concept of performative prediction (Perdomo et al., 2020). In this case, understanding the convergence properties of the algorithm becomes more challenging. Toward this, some recent progress has been made on (strongly) convex stochastic optimization with decision-dependent distributions (Mendler-Dünner et al., 2020; Perdomo et al., 2020; Drusvyatskiy & Xiao, 2023). Moreover, other works have also considered nonconvex problems and obtained some promising results; see (Dong et al., 2023; Jagadeesan et al., 2022) and references therein. Developing theoretical foundation for these problems has become a very active field nowadays.

**RL with general utilities**. It is known that the goal of an agent associated with an MDP is to seek an optimal policy via maximizing the cumulative discounted reward (Sutton & Barto, 2018). However, there are decision problems of interest having more general forms. Beyond the scope of the expected cumulative reward in MDPs, some recent works also looked into RL problems with general utilities; see e.g., (Zhang et al., 2020a; Kumar et al., 2022; Barakat et al., 2023) as mentioned previously. Global convergence results can also be derived via investigating the hidden convex structure (Zhang et al., 2020a) inherited from the MDP.

## 3 Preliminary

In this paper, we assume that the optimal objective value for problem (1), denoted by $\mathcal{F}^*$, is finite and attained, and the reward function $\mathcal{R}_\theta(\cdot)$ satisfies the following assumption.

**Assumption 3.1.** *The following three conditions with respect to the function $\mathcal{R}_\theta(\cdot)$ hold:*

1. *There exists a constant $U > 0$ such that*

$$\sup_{\theta \in \mathbb{R}^n, x \in \mathbb{R}^d} |\mathcal{R}_\theta(x)| \le U.$$

2. *$\mathcal{R}_\theta(\cdot)$ is twice continuously differentiable with respect to $\theta$, and there exist positive constants $\widetilde{C}_g$ and $\widetilde{C}_h$ such that*

$$\sup_{\theta \in \mathbb{R}^n, \ x \in \mathbb{R}^d} \|\nabla_\theta \mathcal{R}_\theta(x)\| \le \widetilde{C}_g, \qquad \sup_{\theta \in \mathbb{R}^n, \ x \in \mathbb{R}^d} \left\|\nabla_\theta^2 \mathcal{R}_\theta(x)\right\|_2 \le \widetilde{C}_h.$$

The first condition on the boundedness of the function $\mathcal{R}_\theta(\cdot)$, which is commonly assumed in the literature (Sutton & Barto, 2018), ensures that $\mathcal{J}(\theta)$ is well-defined. And the second condition will be used to guarantee the well-definiteness and L-smoothness of the gradient $\nabla_\theta \mathcal{J}(\theta)$. We remark here that when the reward function $\mathcal{R}_\theta(x)$ does not depend on $\theta$ (see e.g., Example 3.3), then the second assumption holds automatically.

To determine the (theoretical) learning rate in our algorithmic frameworks, we also need to make some standard assumptions to establish the L-smoothness of $\mathcal{J}(\cdot)$.

**Assumption 3.2** (Lipschitz and smooth policy assumption). *The function $\log \pi_\theta(x)$ is twice differential with respect to $\theta \in \mathbb{R}^n$ and there exist positive constants $C_g$ and $C_h$ such that*

$$\sup_{x \in \mathbb{R}^d, \ \theta \in \mathbb{R}^n} \|\nabla_\theta \log \pi_\theta(x)\| \le C_g, \qquad \sup_{x \in \mathbb{R}^d, \ \theta \in \mathbb{R}^n} \left\|\nabla_\theta^2 \log \pi_\theta(x)\right\|_2 \le C_h.$$

This assumption is a standard one and commonly employed in the literature when studying the convergence properties of the policy gradient method for MDPs; see e.g., (Pirotta et al., 2015; Papini et al., 2018; Xu et al., 2020; Pham et al., 2020; Zhang et al., 2021a; Yang et al., 2022) and references therein.

Under Assumption 3.1 and Assumption 3.2, it is easy to verify that the gradient for the expected reward function $\mathcal{J}(\theta)$ can be written as:

$$\nabla_\theta \mathcal{J}(\theta) = \nabla_\theta \left( \int \mathcal{R}_\theta(x) \pi_\theta(x) dx \right) = \int \left( \nabla_\theta \mathcal{R}_\theta(x) + \mathcal{R}_\theta(x) \frac{\nabla_\theta \pi_\theta(x)}{\pi_\theta(x)} \right) \pi_\theta(x) dx$$

$$= \mathbb{E}_{x \sim \pi_\theta} \left[ \mathcal{R}_\theta(x) \nabla_\theta \log \pi_\theta(x) + \nabla_\theta \mathcal{R}_\theta(x) \right].$$

We next present an example on the discrete-time discounted MDP, which can be covered by the general model (1).

**Example 3.3** (MDP). *We denote a discrete-time discounted MDP as $\mathcal{M} := \{S, A, P, R, \gamma, \rho_0\}$, where $S$ and $A$ denote the state space and the action space, respectively, $P(s'|s,a)$ is the state transition probability from $s$ to $s'$ after selecting the action $a$, $R : S \times A \to [0, U]$ is the reward function that is assumed to be uniformly bounded by a constant $U > 0$, $\gamma \in [0, 1)$ is the discount factor, and $\rho_0$ is the initial state distribution.*

*The agent selects actions according to a stationary random policy $\tilde{\pi}_\theta(\cdot|\cdot) : A \times S \to [0, 1]$ parameterized by $\theta \in \mathbb{R}^n$. Given an initial state $s_0 \in S$, a trajectory $\tau := \{s_t, a_t, r_{t+1}\}_{t=0}^{H-1}$ can then be generated, where $s_0 \sim \rho$, $a_t \sim \tilde{\pi}_\theta(\cdot|s_t)$, $r_{t+1} = R(s_t, a_t)$, $s_{t+1} \sim P(\cdot|s_t, a_t)$, and $H > 0$ is a finite horizon, and the accumulated discounted reward of the trajectory $\tau$ can be defined as $\mathcal{R}(\tau) := \sum_{t=0}^{H-1} \gamma^t r_{t+1}$. Then, the learning objective is to compute an optimal parameter $\theta^*$ that maximizes the expected reward function $\mathcal{J}(\theta)$ [1], i.e.,*

$$\theta^* \in \operatorname{argmax}_\theta \mathcal{J}(\theta) := \mathbb{E}_{\tau \sim \rho_\theta} \left[ \mathcal{R}(\tau) \right], \tag{2}$$

*where $\rho_\theta(\tau) := \rho_0(s_0) \prod_{t=0}^{H-1} P(s_{t+1}|s_t, a_t) \tilde{\pi}_\theta(a_t|s_t)$ denotes the probability distribution of a trajectory $\tau$ being sampled from $\rho_\theta$ that is parameterized by $\theta$.*

*In the special case when $S = \{s\}$ (i.e., $|S| = 1$) and $\gamma = 0$, the MDP reduced to a multi-armed bandit problem (Robbins, 1952) with a reward function simplified as $R : A \to \mathbb{R}$. Particularly, a trajectory $\tau = \{s, a\}$ with the horizon $H_\tau = 0$ is generated, where $a \sim \rho_\theta(\cdot) := \tilde{\pi}_\theta(\cdot|s)$, and the accumulated discounted reward reduces to $\mathcal{R}(x) = R(a)$. As a consequence, problem (2) can be simplified as*

$$\max_{\theta \in \mathbb{R}^n} \mathcal{J}(\theta) = \mathbb{E}_{a \sim \rho_\theta} \left[ R(a) \right].$$

*By adding a convex regularizer $\mathcal{G}(\theta)$ to problem (2), we get the following regularized MDP:*

$$\max_{\theta \in \mathbb{R}^n} \mathbb{E}_{\tau \sim \rho_\theta} \left[ \mathcal{R}(\tau) \right] - \mathcal{G}(\theta),$$

*which was considered in (Pham et al., 2020). However, it is clear that $\mathcal{R}(\tau)$ does not depend on $\theta$. Hence, the above regularized MDP is a special case of the proposed regularized reward optimization problem (1).*

*One can check that the gradient $\nabla_\theta \mathcal{J}(\theta)$ has the following form (Yuan et al., 2022):*

$$\nabla_\theta \mathcal{J}(\theta) = \mathbb{E}_{\tau \sim \rho_\theta} \left[ \sum_{t=0}^{H-1} \gamma^t R(s_t, a_t) \sum_{t'=0}^{t} \nabla_\theta \log \tilde{\pi}_\theta(a_{t'}|s_{t'}) \right].$$

Being a composite optimization problem, problem (1) admits the following first-order stationary condition

$$0 \in -\nabla_\theta \mathcal{J}(\theta) + \partial \mathcal{G}(\theta). \tag{3}$$

Here, $\partial \mathcal{G}(\cdot)$ denotes the subdifferential of the proper closed and convex function $\mathcal{G}(\cdot)$ which is defined as

$$\partial \mathcal{G}(\theta) := \{ g \in \mathbb{R}^n \; : \; \mathcal{G}(\theta') \geq \mathcal{G}(\theta) + \langle g, \theta' - \theta \rangle, \; \forall \theta \}.$$

It is well-known that $\partial \mathcal{G}(\theta)$ is a nonempty closed convex subset of $\mathbb{R}^n$ for any $\theta \in \mathbb{R}^n$ such that $\mathcal{G}(\theta) < \infty$ (see e.g., (Rockafellar, 1997)). Note that any optimal solution of problem (1) satisfies the condition (3), while the reverse statement is generally not valid for nonconcave problems, including the problem (1). The condition (3) leads to the following concept of stationary points for problem (1).

---

[1] Here, the trajectory $\tau$ and the distribution $\rho_\theta$ correspond to $x$ and $\pi_\theta$ in (1), respectively.

**Definition 3.4.** *A point $\theta \in \mathbb{R}^n$ is called a stationary point for problem (1) if it satisfies the condition (3). Given a tolerance $\epsilon > 0$, a stochastic optimization method attains an (expected) $\epsilon$-stationary point, denoted as $\theta \in \mathbb{R}^n$, if*

$$\mathbb{E}_T \left[ \text{dist} \left( 0, -\nabla_\theta \mathcal{J}(\theta) + \partial \mathcal{G}(\theta) \right)^2 \right] \leq \epsilon^2,$$

*where the expectation is taken with respect to all the randomness caused by the algorithm, after running it $T$ iterations, and $\text{dist}(x, \mathcal{C})$ denotes the distance between a point $x$ and a closed convex set $\mathcal{C}$.*

**Remark 3.5** (Gradient mapping). *Note that the optimality condition (3) can be rewritten as*

$$0 = G_\eta(\theta) := \frac{1}{\eta} \left[ \text{Prox}_{\eta \mathcal{G}} \left( \theta + \eta \nabla_\theta \mathcal{J}(\theta) \right) - \theta \right],$$

*for some $\eta > 0$, where*

$$\text{Prox}_{\eta \mathcal{G}}(\theta) := \text{argmin}_{\theta'} \left\{ \mathcal{G}(\theta') + \frac{1}{2\eta} \| \theta' - \theta \|^2 \right\}$$

*denotes the proximal mapping of the function $\mathcal{G}(\cdot)$. The mapping $G_\eta(\cdot)$ is called the gradient mapping in the field of optimization (Beck, 2017). It is easy to verify that if for a $\theta \in \mathbb{R}^n$, it holds that*

$$\text{dist} \left( 0, -\nabla_\theta \mathcal{J}(\theta) + \partial \mathcal{G}(\theta) \right) \leq \epsilon,$$

*then there exists a vector $d$ satisfying $\|d\| \leq \epsilon$ such that*

$$d + \nabla_\theta \mathcal{J}(\theta) \in \partial \mathcal{G}(\theta),$$

*which is equivalent to saying that*

$$\theta = \text{Prox}_{\eta \mathcal{G}} \left( \eta d + \theta + \eta \nabla_\theta \mathcal{J}(\theta) \right).$$

*Moreover, we can verify that (by using the firm nonexpansiveness of $\text{Prox}_{\eta \mathcal{G}}(\cdot)$; see e.g., (Beck, 2017))*

$$\| G_\eta(\theta) \| = \frac{1}{\eta} \| \text{Prox}_{\eta \mathcal{G}} \left( \theta + \eta \nabla_\theta \mathcal{J}(\theta) \right) - \theta \| \leq \|d\| \leq \epsilon.$$

*Therefore, we can also characterize an (expected) $\epsilon$-stationary point by using the following condition*

$$\mathbb{E}_T \left[ \| G_\eta(\theta) \|^2 \right] \leq \epsilon^2.$$

The main objective of this paper is to study the convergence properties, including iteration and sample complexities, of the stochastic (variance-reduced) proximal gradient method to a $\epsilon$-stationary point with a pre-specified $\epsilon > 0$. Note that all proofs of our results are presented in the appendix. Moreover, we acknowledge that our analysis is drawn upon classical results in the literature.

## 4   The stochastic proximal gradient method

In this section, we present and analyze the stochastic proximal gradient method for solving the problem (1). The fundamental idea of the algorithm is to replace the true gradient $\nabla_\theta \mathcal{J}(\theta)$, which are not available for most of the time, with a stochastic gradient estimator in the classical proximal gradient method (Beck, 2017). The method can be viewed as extensions to the projected policy gradient method with direct parameterization (Agarwal et al., 2021) and the stochastic policy gradient method for unregularized MDPs (Williams, 1992). The detailed description of the algorithm is presented in Algorithm 1.

For notational simplicity, we denote

$$g(x, \theta) := \mathcal{R}_\theta(x) \nabla_\theta \log \pi_\theta(x) + \nabla_\theta \mathcal{R}_\theta(x).$$

From Algorithm 1, we see that at each iteration, $N$ data points, namely $\{x^{t,1}, \ldots, x^{t,N}\}$, are sample according to the current probability distribution $\pi_{\theta^t}$. Using these data points, we can construct a REINFORCE-type

---

**Algorithm 1** The stochastic proximal gradient method

---
1: **Input:** initial point $\theta^0$, sample size $N$ and the learning rate $\eta > 0$.
2: **for** $t = 0, \ldots, T-1$ **do**
3:     Compute the stochastic gradient estimator:

$$g^t := \frac{1}{N} \sum_{j=1}^{N} g(x^{t,j}, \theta^t),$$

   where $\{x^{t,1}, \ldots, x^{t,N}\}$ are sampled independently according to $\pi_{\theta^t}$.
4:     Update
$$\theta^{t+1} = \text{Prox}_{\eta\mathcal{G}}\left(\theta^t + \eta g^t\right).$$

5: **end for**
6: **Output:** $\hat{\theta}^T$ selected randomly from the generated sequence $\{\theta^t\}_{t=1}^{T}$.

---

stochastic gradient estimator $g^t$. Then, the algorithm just performs a proximal gradient ascent updating. Let $T > 0$ be the maximal number of iterations, then a sequence $\{\theta^t\}_{t=1}^{T}$ can be generated, and the output solution is selected randomly from this sequence. Next, we shall proceed to answer the questions that how to choose the learning rate $\eta > 0$, how large the sample size $N$ should be, and how many iterations for the algorithm to output an $\epsilon$-stationary point for a given $\epsilon > 0$, theoretically. The next lemma establishes the L-smoothness of $\mathcal{J}(\cdot)$ whose proof is given at Appendix A.1.

**Lemma 4.1.** *Under Assumptions 3.1 and 3.2, the gradient of $\mathcal{J}$ is L-smooth, i.e.,*

$$\|\nabla_\theta \mathcal{J}(\theta) - \nabla_\theta \mathcal{J}(\theta')\| \le L \|\theta - \theta'\|, \quad \forall \theta, \theta' \in \mathbb{R}^n,$$

*with $L := U(C_g^2 + C_h) + \widetilde{C}_h + 2C_g\widetilde{C}_g > 0$.*

**Remark 4.2** (L-smoothness in MDPs)**.** *For an MDP with finite action space and state space as in Example 3.3, the Lipschitz constant of $\nabla_\theta \mathcal{J}(\cdot)$ can be expressed in terms of $|A|$, $|S|$ and $\gamma$. We refer the reader to (Agarwal et al., 2021; Xiao, 2022) for more details.*

As a consequence of the L-smoothness of the function $\mathcal{J}(\cdot)$, we next show that the learning rate can be chosen as a positive constant upper bounded by a constant depends only on the Lipschitz constant of $\nabla_\theta \mathcal{J}(\cdot)$. For notational complicity, we denote $\Delta := \mathcal{F}^* - \mathcal{F}(\theta^0) > 0$ for the rest of this paper.

**Theorem 4.3.** *Under Assumptions 3.1 and 3.2, if we set $\eta \in \left(0, \frac{1}{2L}\right)$, then Algorithm 1 outputs a point $\hat{\theta}^T$ satisfying*

$$\mathbb{E}_T \left[ \text{dist}\left(0, -\nabla_\theta \mathcal{J}(\hat{\theta}^T) + \partial\mathcal{G}(\hat{\theta}^T)\right)^2 \right]$$

$$\le \left(2 + \frac{2}{\eta L(1 - 2\eta L)}\right) \frac{1}{T} \sum_{t=0}^{T-1} \mathbb{E}_T \left[\|g^t - \nabla_\theta \mathcal{J}(\theta^t)\|^2\right] + \frac{\Delta}{T}\left(\frac{2}{\eta} + \frac{4}{\eta(1 - 2\eta L)}\right),$$

*where $\mathbb{E}_T$ is defined in Definition 3.4.*

The proof of the above theorem is provided in Appendix A.2. From this theorem, if one sets $g^t = \nabla_\theta \mathcal{J}(\theta^t)$, i.e., $\|g^t - \nabla_\theta \mathcal{J}(\theta^t)\|^2 = 0$, then there is no randomness along the iterations and the convergence property is reduced to

$$\min_{1 \le t \le T} \text{dist}\left(0, -\nabla_\theta \mathcal{J}(\hat{\theta}^t) + \partial\mathcal{G}(\hat{\theta}^t)\right) = O\left(\frac{1}{\sqrt{T}}\right),$$

which is implied by classical results on proximal gradient method (see e.g., (Beck, 2017)). However, since the exact full gradient $\nabla_\theta \mathcal{J}(\theta)$ is rarely computable, it is common to require the variance (i.e., the trace of the covariance matrix) of the stochastic estimator to be bounded. The latter condition plays an essential role in analyzing stochastic first-order methods for solving nonconvex optimization problems, including RL applications; see, e.g., (Beck, 2017; Papini et al., 2018; Shen et al., 2019; Lan, 2020; Yang et al., 2022).

**Lemma 4.4.** *Under Assumptions 3.1 and 3.2, there exists a constant $\sigma > 0$ such that for any $\theta$,*

$$\mathbb{E}_{x \sim \pi_\theta} \left[ \|g(x, \theta) - \nabla_\theta \mathcal{J}(\theta)\|^2 \right] \leq \sigma^2.$$

The proof of Lemma 4.4 is given in Appendix A.3. By choosing a suitable sample size $N$, we can rely on Lemma 4.4 to make the term $\mathbb{E}_T \left[ \|g^t - \nabla_\theta \mathcal{J}(\theta^t)\|^2 \right]$ in Theorem 4.3 small, for every $t \leq T$. Then, Theorem 4.3 implies that Algorithm 1 admits an expected $O(T^{-1})$ convergence rate to a stationary point. These results are summarized in the following theorem; see Appendix A.4 for a proof.

**Theorem 4.5.** *Suppose that Assumptions 3.1 and 3.2 hold. Let $\epsilon > 0$ be a given accuracy. Running the Algorithm 1 for*

$$T := \left\lceil \frac{\Delta}{\epsilon^2} \left( \frac{4}{\eta} + \frac{8}{\eta(1 - 2\eta L)} \right) \right\rceil = O(\epsilon^{-2})$$

*iterations with the learning rate $\eta < \frac{1}{2L}$ and the sample size*

$$N := \left\lceil \frac{\sigma^2}{\epsilon^2} \left( 4 + \frac{4}{\eta L(1 - 2\eta L)} \right) \right\rceil = O(\epsilon^{-2})$$

*outputs a point $\hat{\theta}^T$ satisfying*

$$\mathbb{E}_T \left[ \text{dist} \left( 0, -\nabla_\theta \mathcal{J}(\hat{\theta}^T) + \partial \mathcal{G}(\hat{\theta}^T) \right)^2 \right] \leq \epsilon^2.$$

*Moreover, the sample complexity is $O(\epsilon^{-4})$.*

As already mentioned in the introduction, the total sample complexity of Algorithm 1 to an $\epsilon$-stationary point is shown to be $O(\epsilon^{-4})$, which matches the most competitive sample complexity of the classical stochastic policy gradient for MDPs (Williams, 1992; Baxter & Bartlett, 2001; Zhang et al., 2020b; Xiong et al., 2021; Yuan et al., 2022).

**Remark 4.6** (Sample size). *Note that the current state-of-the-art iteration complexity for the (small-batch) stochastic gradient descent method is $T := O(\epsilon^{-2})$ with $\eta_t := \min\{O(L^{-1}), O(T^{-1/2})\}$; see, e.g., (Ghadimi & Lan, 2013). The reason for requiring larger batch-size in Theorem 4.5 is to allow a constant learning rate. To the best of our knowledge, to get the same convergence properties as Theorem 4.5 under the same conditions for problem (1), the large batch-size is required.*

**Remark 4.7** (Global convergence). *As mentioned in introduction, some recent progress has been made for analyzing the global convergence properties of the policy gradient methods for MDPs, which greatly rely on the concepts of gradient domination and its extensions (Agarwal et al., 2021; Mei et al., 2020; Xiao, 2022; Yuan et al., 2022; Gargiani et al., 2022). This concept is also highly related to the classical PŁ-condition (Polyak, 1963) and KŁ-condition (Bolte et al., 2007) in the field of optimization. One of the key ideas is to assume or verify that the difference between the optimal objective function value, namely $\mathcal{F}^*$, and $\mathcal{F}(\theta)$ can be bounded by the quantity depending on the norm of the gradient mapping at an arbitrary point. In particular, suppose that there exists a positive constant $\omega$ such that*

$$\|G_\eta(\theta))\| \geq 2\sqrt{\omega} \left( \mathcal{F}^* - \mathcal{F}(\theta) \right), \quad \forall \theta \in \mathbb{R}^n,$$

*where $G_\eta$ is defined in Remark 3.5 (see e.g., (Xiao, 2022)). Then, after running Algorithm 2 for $T = O(\epsilon^{-2})$ iterations, one can easily check that*

$$\mathbb{E}_T \left[ \mathcal{F}^* - \mathcal{F}(\hat{\theta}^T) \right] \leq \frac{1}{2\sqrt{\omega}} \epsilon.$$

*As a conclusion, by assuming or verifying stronger conditions, one can typically show that any stationary point of the problem (1) is also a globally optimal solution. This shares the same spirit of (Zhang et al., 2020a) for MDPs with general utilities. We leave it as a future research to analyze the global convergence of the problem (1).*

## 5 Variance reduction via PAGE

Recall from Theorem 4.3 that, there is a trade-off between the sample complexity and the iteration complexity of Algorithm 1. In particular, while there is little room for us to improve the term $\frac{\Delta}{T}\left(\frac{2}{\eta} + \frac{4}{\eta(1-2\eta L)}\right)$ which corresponds to the iteration complexity, it is possible to construct $g^t$ in an advanced manner to improve the sample complexity. Therefore, our main goal in this section is to reduce the expected sample complexity while keeping the term $\frac{1}{T}\sum_{t=0}^{T-1}\mathbb{E}_T\left[\left\|\nabla_\theta \mathcal{J}(\theta^t) - g^t\right\|^2\right]$ small. We achieve this goal by considering the stochastic variance-reduced gradient methods that have recently attracted much attention. Among these variance-reduced methods, as argued in (Gargiani et al., 2022), the ProbAbilistic Gradient Estimator (PAGE) proposed in (Li et al., 2021b) has a simple structure, and can lead to optimal convergence properties. These appealing features make it attractive in machine learning applications. Therefore, in this section, we also consider the stochastic variance-reduced proximal gradient method with PAGE for solving the problem (1).

PAGE is originally designed for the stochastic nonconvex minimization in the oblivious setting:

$$\min_{\theta \in \mathbb{R}^n} \ f(\theta) := \mathbb{E}_{x \sim \pi}[F(x, \theta)]$$

where $\pi$ is a fixed probability distribution and $F : \mathbb{R}^d \times \mathbb{R}^n \to \mathbb{R}$ is a certain differentiable (and possibly nonconvex) loss function. For stochastic gradient-type methods, a certain stochastic gradient estimator for $f$ is required for performing the optimization. At the $t$-th iteration, given a probability $p_t \in [0, 1]$ and the current gradient estimator $g^t$, PAGE proposed to replace the vanilla mini-batch gradient estimator with the following unbiased stochastic estimator:

$$\nabla f(\theta^{t+1}) \approx g^{t+1} := \begin{cases} \dfrac{1}{N_1}\sum_{j=1}^{N_1}\nabla_\theta F(x^j, \theta^{t+1}), & \text{with probability } p_t, \\ g^t + \dfrac{1}{N_2}\left(\sum_{j=1}^{N_2}\nabla_\theta F(x^j, \theta^{t+1}) - \sum_{j=1}^{N_2}\nabla_\theta F(x^j, \theta^t)\right), & \text{with probability } 1 - p_t, \end{cases}$$

where $\{x^j\}$ are sampled from $\pi$, $N_1, N_2$ denote the sample sizes. Some key advantages of applying PAGE are summarized as follows. First, the algorithm is single-looped, which admit simpler implementation compared with existing double-looped variance reduced methods. Second, the probability $p_t$ can be adjusted dynamically, leading to more flexibilities. Third, one can choose $N_2$ to be much smaller than $N_1$ to guarantee the same iteration complexity as the vanilla SGD. Thus, the overall sample complexity can be significantly reduced. However, the application of PAGE to our setting needs significant modifications and extensions, which we shall demonstrate below. To the best of our knowledge, the application of PAGE for solving the general regularized reward optimization problem in the non-oblivious setting considered in this paper is new.

For notational simplicity, for the rest of this section, we denote

$$g_w(x, \theta, \theta') = \frac{\pi_\theta(x)}{\pi_{\theta'}(x)}g(x, \theta),$$

for $\theta, \theta' \in \mathbb{R}^n$, $x \in \mathbb{R}^d$, where $\frac{\pi_\theta(x)}{\pi_{\theta'}(x)}$ denotes the importance weight between $\pi_\theta$ and $\pi_{\theta'}$. Note also that

$$\mathbb{E}_{x \sim \pi_{\theta'}}\left[\frac{\pi_\theta(x)}{\pi_{\theta'}(x)}\right] = 1.$$

The description of the proposed PAGE variance-reduced stochastic proximal gradient method is given in Algorithm 2.

It is clear that the only difference between Algorithm 1 and Algorithm 2 is the choice of the gradient estimator. At each iteration of the latter algorithm, we have two choices for the gradient estimator, where, with probability $p$, one chooses the same estimator as in Algorithm 1 with a sample size $N_1$, and with

---

**Algorithm 2** The variance-reduced stochastic proximal gradient method with PAGE

---

1: **Input:** initial point $\theta^0$, sample sizes $N_1$ and $N_2$, a probability $p \in (0,1]$, and the learning rate $\eta > 0$.
2: Compute

$$g^0 := \frac{1}{N_1} \sum_{j=1}^{N_1} g(x^{0,j}, \theta^0),$$

where $\{x^{0,j}\}_j$ are sampled independently according to $\pi_{\theta^0}$.
3: **for** $t = 0, \ldots, T-1$ **do**
4:    Update

$$\theta^{t+1} = \text{Prox}_{\eta\mathcal{G}} \left( \theta^t + \eta g^t \right).$$

5:    Compute

$$g^{t+1} = \begin{cases} \dfrac{1}{N_1} \sum_{j=1}^{N_1} g(x^{t+1,j}, \theta^{t+1}), & \text{with probability } p, \\ \dfrac{1}{N_2} \sum_{j=1}^{N_2} g(x^{t+1,j}, \theta^{t+1}) - \dfrac{1}{N_2} \sum_{j=1}^{N_2} g_w(x^{t+1,j}, \theta^t, \theta^{t+1}) + g^t, & \text{with probability } 1-p, \end{cases}$$

where $\{x^{t+1,j}\}_j$ are sampled independently according to $\pi_{\theta^{t+1}}$.
6: **end for**
7: **Output:** $\hat{\theta}^T$ selected randomly from the generated sequence $\{\theta^t\}_{t=1}^T$.

---

probability $1-p$, one constructs the estimator in a clever way which combines the information of the current iterate and the previous one. Since the data set $\{x^{t+1,1}, \ldots, x^{t+1,N_2}\}$ is sampled according to the current probability distribution $\pi_{\theta^{t+1}}$, we need to rely on the importance weight between $\theta^t$ and $\theta^{t+1}$ and construct the gradient estimator $\dfrac{1}{N_2} \sum_{j=1}^{N_2} g_w(x^{t+1}, \theta^t, \theta^{t+1})$, which is an unbiased estimator for $\nabla_\theta \mathcal{J}(\theta^t)$, so that $g^{t+1}$ becomes an unbiased estimator of $\nabla_\theta \mathcal{J}(\theta^{t+1})$. Indeed, one can easily verify that for any $\theta, \theta' \in \mathbb{R}^n$, it holds that

$$\mathbb{E}_{x \sim \pi_{\theta'}} [g_w(x, \theta, \theta')] = \nabla_\theta \mathcal{J}(\theta), \tag{4}$$

i.e., $g(x, \theta, \theta')$ is an unbiased estimator for $\nabla_\theta \mathcal{J}(\theta)$ provided that $x \sim \pi_{\theta'}$.

Next, we shall analyze the convergence properties of Algorithm 2. Our analysis relies on the following assumption on the importance weight, which essentially controls the change of the distributions.

**Assumption 5.1.** *Let $\theta, \theta' \in \mathbb{R}^n$, the importance weight between $\pi_\theta$ and $\pi_{\theta'}$ is well-defined and there exists a constant $C_w > 0$ such that*

$$\mathbb{E}_{x \sim \pi_{\theta'}} \left[ \left( \frac{\pi_\theta(x)}{\pi_{\theta'}(x)} - 1 \right)^2 \right] \leq C_w^2.$$

Clearly, the significance of the constant $C_w$ (if exists) may depend sensitively on $\theta$ and $\theta'$. To see this, let us assume that for any $\theta \in \mathbb{R}^n$, $\pi_\theta = \theta$ is a discrete distribution over a set of finite points $\{x_k\}_{k=1}^n$ for which $\pi_\theta(x_k) = \theta_k > 0$ for all $k = 1, \ldots, n$. Now, suppose that $\theta = \theta' + \Delta\theta$ with $|\Delta\theta_k| \leq 1$. Then, a simple calculation shows that

$$\mathbb{E}_{x \sim \pi_{\theta'}} \left[ \left( \frac{\pi_\theta(x)}{\pi_{\theta'}(x)} - 1 \right)^2 \right] = \sum_{k=1}^n \left( \frac{\theta_k}{\theta'_k} - 1 \right)^2 \theta'_k = \sum_{k=1}^n \frac{\theta_k \Delta\theta_k}{\theta'_k} \leq \sum_{k=1}^n \frac{\theta_k}{\theta'_k}.$$

However, it is possible that there exists a certain $\theta'_k = 0$ or tiny. In this case, $C_w$ can be huge or even infinity. Fortunately, the regularization term $\mathcal{G}(\theta)$ can help to avoid such undesired situations via imposing the lower-bounded constraints $\theta_k \geq \delta > 0$ for all $k$. In this case, we see that $\sum_{k=1}^n \frac{\theta_k}{\theta'_k} \leq \sum_{k=1}^n \frac{\theta_k}{\delta} = \frac{1}{\delta}$.

**Remark 5.2.** *Note that Assumption 5.1 is also employed in many existing works (Papini et al., 2018; Xu et al., 2019; Pham et al., 2020; Yuan et al., 2020; Gargiani et al., 2022). However, this assumption could be too strong, and it is not checkable in general. Addressing the relaxation of this assumption through the development of a more sophisticated algorithmic framework is beyond the scope of this paper. Here, we would like to mention some recent progress on relaxing this stringent condition for MDPs. By constructing additional stochastic estimators for the Hessian matrix of the objective function, (Shen et al., 2019) proposed a Hessian-aided policy-gradient-type method that improves the sample complexity from $O(\epsilon^{-4})$ to $O(\epsilon^{-3})$ without assuming Assumption 5.1. Later, by explicitly controlling changes in the parameter $\theta$, (Zhang et al., 2021a) developed a truncated stochastic incremental variance-reduced policy gradient method to prevent the variance of the importance weights from becoming excessively large leading to the $O(\epsilon^{-3})$ sample complexity. By utilizing general Bregman divergences, (Yuan et al., 2022) proposed a double-looped variance-reduced mirror policy optimization approach and established an $O(\epsilon^{-3})$ sample complexity, without requiring Hessian information or Assumption 5.1. Recently, following the research theme as (Shen et al., 2019), (Salehkaleybar et al., 2022) also incorporated second-order information into the stochastic gradient estimator. By using momentum, the variance-reduced algorithm proposed in (Salehkaleybar et al., 2022) has some appealing features, including the small batch-size and parameter-free implementation. Recently, by imposing additional conditions, including the Lipschitz continuity of the Hessian of the score function $\nabla_\theta \log \pi_\theta$ and the Fisher-non-degeneracy condition of the policy, (Fatkhullin et al., 2023) derived improved (global) convergence guarantees for solving MDPs. We think that the above ideas can also be explored for solving the general model (1).*

The bounded variance of the importance weight implies that the (expected) distance between $g(x, \theta')$ and $g_w(x, \theta, \theta')$ is controlled by the distance between $\theta$ and $\theta'$, for any given $\theta, \theta' \in \mathbb{R}^d$. In particular, we have the following lemma, whose proof is provided in Appendix A.5.

**Lemma 5.3.** *Under Assumption 3.1, Assumption 3.2 and Assumption 5.1, then it holds that*

$$\mathbb{E}_{x \sim \pi_{\theta'}} \left[ \|g(x, \theta') - g_w(x, \theta, \theta')\|^2 \right] \leq C \|\theta - \theta'\|^2,$$

*where $C > 0$ is a constant defined as*

$$C := 6U^2 C_h^2 + 6C_g^2 \widetilde{C}_g^2 + 6\widetilde{C}_h^2 + \left( 4U^2 C_g^2 + 4\widetilde{C}_g^2 \right) (2C_g^2 + C_h)(C_w^2 + 1).$$

Under the considered assumptions, we are able to provide an estimate for the term $\sum_{t=0}^{T-1} \mathbb{E}_T \left[ \|g^t - \nabla_\theta \mathcal{J}(\theta^t)\|^2 \right]$, which plays an essential role in deriving an improved sample complexity of Algorithm 2. The results are summarized in the following Lemma 5.4; see Appendix A.6 for a proof which shares the same spirit as (Li et al., 2021b, Lemma 3 & 4).

**Lemma 5.4.** *Suppose that Assumption 3.1, Assumption 3.2, and Assumption 5.1 hold. Let $\{g^t\}$ and $\{\theta^t\}$ be the sequences generated by Algorithm 2, then it holds that*

$$\left( 1 - \frac{(1-p)C\eta}{pN_2 L(1-2\eta L)} \right) \sum_{t=0}^{T-1} \mathbb{E}_T \left[ \|g^t - \nabla_\theta \mathcal{J}(\theta^t)\|^2 \right] \leq \frac{p\sigma^2 T + \sigma^2}{pN_1} + \frac{2\eta(1-p)C\Delta}{pN_2(1-2\eta L)}.$$

We are now ready to present the main result on the convergence property of the Algorithm 2 by showing how to select the sample sizes $N_1$ and $N_2$, probability $p$, and the learning rate $\eta$. Intuitively, $N_1$ is typically a large number and one does not want to perform samplings with $N_1$ samples frequently, thus the probability $p$ and the sample size $N_2$ should both be small. Given $N_1$, $N_2$ and $p$, we can then determine the value of $\eta$ such that $\eta < \frac{1}{2L}$. Consequently, the key estimate in Theorem 4.3 can be applied directly. Our results are summarized in the following theorem. Reader is referred to Appendix A.7 for the proof of this result.

**Theorem 5.5.** *Suppose that Assumption 3.1, Assumption 3.2 and Assumption 5.1 hold. For a given $\epsilon \in (0, 1)$, we set $p := \frac{N_2}{N_1 + N_2}$ with $N_1 := O(\epsilon^{-2})$ and $N_2 := \sqrt{N_1} = O(\epsilon^{-1})$. Choose a learning rate $\eta$ satisfying*

$\eta \in \left(0, L/(2C + 2L^2)\right]$. *Then, running Algorithm 2 for* $T := O(\epsilon^{-2})$ *iterations outputs a point* $\hat{\theta}^T$ *satisfying*

$$\mathbb{E}_T \left[ \mathrm{dist} \left(0, -\nabla_\theta \mathcal{J}(\hat{\theta}^T) + \partial \mathcal{G}(\hat{\theta}^T)\right)^2 \right] \leq \epsilon^2.$$

*Moreover, the total expected sample complexity is* $O(\epsilon^{-3})$.

By using the stochastic variance-reduce gradient estimator with PAGE and the importance sampling technique, we have improved the total sample complexity from $O(\epsilon^{-4})$ to $O(\epsilon^{-3})$, under the considered conditions. This result matches with the current competitive results established in (Xu et al., 2019; Yuan et al., 2020; Pham et al., 2020; Gargiani et al., 2022) for solving MDPs and is applicable to the general model (1). Finally, as mentioned in Remark 4.7, by assuming or verifying stronger conditions, such as the gradient domination and its extensions, it is also possible to derive some global convergence results. Again, such a possibility is left to a future research direction.

## 6 Conclusions

We have studied the stochastic (variance-reduced) proximal gradient method addressing a general regularized expected reward optimization problem which covers many existing important problem in reinforcement learning. We have established the $O(\epsilon^{-4})$ sample complexity of the classical stochastic proximal gradient method and the $O(\epsilon^{-3})$ sample complexity of the stochastic variance-reduced proximal gradient method with an importance sampling based probabilistic gradient estimator. Our results match the sample complexity of their most competitive counterparts under similar settings for Markov decision processes.

Meanwhile, we have also suspected some limitations in the current paper. First, due to the nonconcavity of the objective function, we found it challenging to derive global convergence properties of the stochastic proximal gradient method and its variants without imposing additional conditions. On the other hand, analyzing the sample complexity for achieving convergence to second-order stationary points—thereby avoiding saddle points—may be more realistic and feasible (Arjevani et al., 2020). Second, the bounded variance condition for the importance weight turns out to be quite strong and can not be verified in general. How to relax this condition for our general model deserves further investigation. Last but not least, since we focus more on the theoretical analysis in this paper and due to the space constraint, we did not conduct any numerical simulation to examine the practical efficiency of the proposed methods. We shall try to delve into these challenges and get better understandings of the proposed problem and algorithms in a future research.

Finally, this paper has demonstrated the possibility of pairing the stochastic proximal gradient method with efficient variance reduction techniques (Li et al., 2021b) for solving the reward optimization problem (1). Beyond variance-reduced methods, there are other possibilities that allow one deriving more sophisticated algorithms. For instance, one can also pair the stochastic proximal gradient method with the ideas of the actor-critic method (Konda & Tsitsiklis, 1999), the natural policy gradient method (Kakade, 2001), policy mirror descent methods (Tomar et al., 2020; Lan, 2023), trust-region methods (Schulman et al., 2015; Shani et al., 2020), and the variational policy gradient methods (Zhang et al., 2020a). We think that these possible generalizations can lead to more exciting results and make further contributions to the literature.

### Acknowledgments

We thank the action editor and reviewers for their valuable comments and suggestions that helped to improve the quality of the paper. The authors were partially supported by the US National Science Foundation under awards DMS-2244988, DMS-2206333, and the Office of Naval Research Award N00014-23-1-2007.

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

# A  Proofs

## A.1  Proof of Lemma 4.1

*Proof of Lemma 4.1.* One could establish the $L$-smoothness of $\mathcal{J}(\cdot)$ via bounding the spectral norm of the Hessian $\nabla_\theta^2 \mathcal{J}(\cdot)$. To this end, we first calculate the Hessian of $\mathcal{J}$ as follows:

$$\nabla_\theta^2 \mathcal{J}(\theta) = \nabla_\theta \mathbb{E}_{x \sim \pi_\theta} \left[ \mathcal{R}_\theta(x) \nabla_\theta \log \pi_\theta(x) + \nabla_\theta \mathcal{R}_\theta(x) \right]$$

$$= \nabla_\theta \int \left( \mathcal{R}_\theta(x) \nabla_\theta \log \pi_\theta(x) \pi_\theta(x) + \nabla_\theta \mathcal{R}_\theta(x) \pi_\theta(x) \right) \mathrm{d}x$$

$$= \int \mathcal{R}_\theta(x)\pi_\theta(x) \left( \nabla_\theta^2 \log \pi_\theta(x) + \nabla_\theta \log \pi_\theta(x) \nabla_\theta \log \pi_\theta(x)^\top \right) dx$$

$$+ \int \nabla_\theta^2 \mathcal{R}_\theta(x)\pi_\theta(x) + 2\nabla_\theta \mathcal{R}_\theta(x)\nabla_\theta \pi_\theta(x)^\top dx$$

$$= \mathbb{E}_{x \sim \pi_\theta} \left[ \mathcal{R}_\theta(x)\nabla_\theta^2 \log \pi_\theta(x) \right] + \mathbb{E}_{x \sim \pi_\theta} \left[ \mathcal{R}_\theta(x)\nabla_\theta \log \pi_\theta(x) \nabla_\theta \log \pi_\theta(x)^\top \right]$$

$$+ \mathbb{E}_{x \sim \pi_\theta} \left[ \nabla_\theta^2 \mathcal{R}_\theta(x) \right] + 2\mathbb{E}_{x \sim \pi_\theta} \left[ \nabla_\theta \mathcal{R}_\theta(x)\nabla_\theta \log \pi_\theta(x)^\top \right].$$

Then, by the triangular inequality, it holds that

$$\left\| \nabla_\theta^2 \mathcal{J}(\theta) \right\|_2 \leq \sup_{x \in \mathbb{R}^d, \ \theta \in \mathbb{R}^n} \left\| \mathcal{R}_\theta(x)\nabla_\theta^2 \log \pi_\theta(x) \right\|_2 + \sup_{x \in \mathbb{R}^d, \ \theta \in \mathbb{R}^n} \left\| \mathcal{R}_\theta(x)\nabla_\theta \log \pi_\theta(x)\nabla_\theta \log \pi_\theta(x)^\top \right\|_2$$

$$+ \sup_{x \in \mathbb{R}^d, \ \theta \in \mathbb{R}^n} \left\| \nabla_\theta^2 \mathcal{R}_\theta(x) \right\|_2 + 2 \sup_{x \in \mathbb{R}^d, \ \theta \in \mathbb{R}^n} \left\| \nabla_\theta \mathcal{R}_\theta(x)\nabla_\theta \log \pi_\theta(x)^\top \right\|_2$$

$$\leq U(C_g^2 + C_h) + \widetilde{C}_h + 2C_g\widetilde{C}_g.$$

Thus, $\mathcal{J}$ is $L$-smooth with $L := U(C_g^2 + C_h) + \widetilde{C}_h + 2C_g\widetilde{C}_g$, and the proof is completed. $\qquad \square$

## A.2  Proof of Theorem 4.3

*Proof of Theorem 4.3.* From Lemma 4.1, we see that

$$\mathcal{J}(\theta^{t+1}) \geq \mathcal{J}(\theta^t) + \left\langle \nabla_\theta \mathcal{J}(\theta^t), \theta^{t+1} - \theta^t \right\rangle - \frac{L}{2} \left\| \theta^{t+1} - \theta^t \right\|^2. \tag{5}$$

By the updating rule of $\theta^{t+1}$, we see that

$$- \left\langle g^t, \theta^{t+1} - \theta^t \right\rangle + \frac{1}{2\eta} \left\| \theta^{t+1} - \theta^t \right\|^2 + \mathcal{G}(\theta^{t+1}) \leq \mathcal{G}(\theta^t), \tag{6}$$

$$g^t - \frac{1}{\eta} \left( \theta^{t+1} - \theta^t \right) \in \partial \mathcal{G}(\theta^{t+1}). \tag{7}$$

Combining (5) and (6), we see that

$$\mathcal{J}(\theta^{t+1}) + \left\langle g^t, \theta^{t+1} - \theta^t \right\rangle - \frac{1}{2\eta} \left\| \theta^{t+1} - \theta^t \right\|^2 - \mathcal{G}(\theta^{t+1})$$

$$\geq \mathcal{J}(\theta^t) + \left\langle \nabla_\theta \mathcal{J}(\theta^t), \theta^{t+1} - \theta^t \right\rangle - \frac{L}{2} \left\| \theta^{t+1} - \theta^t \right\|^2 - \mathcal{G}(\theta^t).$$

Rearranging terms, we can rewrite the above inequality as

$$\frac{1 - \eta L}{2\eta} \left\| \theta^{t+1} - \theta^t \right\|^2 \leq \mathcal{F}(\theta^{t+1}) - \mathcal{F}(\theta^t) + \left\langle g^t - \nabla_\theta \mathcal{J}(\theta^t), \theta^{t+1} - \theta^t \right\rangle. \tag{8}$$

By the Cauchy-Schwarz inequality, we see that

$$\left\langle g^t - \nabla_\theta \mathcal{J}(\theta^t), \theta^{t+1} - \theta^t \right\rangle \leq \frac{1}{2L} \left\| g^t - \nabla_\theta \mathcal{J}(\theta^t) \right\|^2 + \frac{L}{2} \left\| \theta^{t+1} - \theta^t \right\|^2,$$

which together with (8) implies that

$$\frac{1 - 2\eta L}{2\eta} \left\| \theta^{t+1} - \theta^t \right\|^2 \leq \mathcal{F}(\theta^{t+1}) - \mathcal{F}(\theta^t) + \frac{1}{2L} \left\| g^t - \nabla_\theta \mathcal{J}(\theta^t) \right\|^2.$$

Summing the above inequality across $t = 0, \ldots, T - 1$, we get

$$\frac{1 - 2\eta L}{2\eta} \sum_{t=0}^{T-1} \left\| \theta^{t+1} - \theta^t \right\|^2 \leq \mathcal{F}(\theta^T) - \mathcal{F}(\theta^0) + \frac{1}{2L} \sum_{t=0}^{T-1} \left\| g^t - \nabla_\theta \mathcal{J}(\theta^t) \right\|^2$$

$$\leq \Delta + \frac{1}{2L} \sum_{t=0}^{T-1} \left\| g^t - \nabla_\theta \mathcal{J}(\theta^t) \right\|^2. \tag{9}$$

Here, we recall that $\Delta := \mathcal{F}^* - \mathcal{F}(\theta^0) > 0$.

On the other hand, (8) also implies that

$$2 \left\langle \nabla_\theta \mathcal{J}(\theta^{t+1}) - g^t, \frac{1}{\eta} \left( \theta^{t+1} - \theta^t \right) \right\rangle + \frac{1 - \eta L}{\eta^2} \left\| \theta^{t+1} - \theta^t \right\|^2$$

$$\leq \frac{2}{\eta} \left( \mathcal{F}(\theta^{t+1}) - \mathcal{F}(\theta^t) \right) + \frac{2}{\eta} \left\langle \nabla_\theta \mathcal{J}(\theta^{t+1}) - \nabla_\theta \mathcal{J}(\theta^t), \theta^{t+1} - \theta^t \right\rangle. \tag{10}$$

Notice that

$$2 \left\langle \nabla_\theta \mathcal{J}(\theta^{t+1}) - g^t, \frac{1}{\eta} \left( \theta^{t+1} - \theta^t \right) \right\rangle$$

$$= \left\| \nabla_\theta \mathcal{J}(\theta^{t+1}) - g^t + \frac{1}{\eta} \left( \theta^{t+1} - \theta^t \right) \right\|^2 - \left\| \nabla_\theta \mathcal{J}(\theta^{t+1}) - g^t \right\|^2 - \frac{1}{\eta^2} \left\| \theta^{t+1} - \theta^t \right\|^2.$$

Then by substituting the above equality into (10) and rearranging terms, we see that

$$\left\| \nabla_\theta \mathcal{J}(\theta^{t+1}) - g^t + \frac{1}{\eta} \left( \theta^{t+1} - \theta^t \right) \right\|^2$$

$$\leq \left\| \nabla_\theta \mathcal{J}(\theta^{t+1}) - g^t \right\|^2 + \frac{1}{\eta^2} \left\| \theta^{t+1} - \theta^t \right\|^2 - \frac{1 - \eta L}{\eta^2} \left\| \theta^{t+1} - \theta^t \right\|^2$$

$$+ \frac{2}{\eta} \left( \mathcal{F}(\theta^{t+1}) - \mathcal{F}(\theta^t) \right) + \frac{2}{\eta} \left\langle \nabla_\theta \mathcal{J}(\theta^{t+1}) - \nabla_\theta \mathcal{J}(\theta^t), \theta^{t+1} - \theta^t \right\rangle$$

$$\leq 2 \left\| \nabla_\theta \mathcal{J}(\theta^t) - g^t \right\|^2 + 2 \left\| \nabla_\theta \mathcal{J}(\theta^{t+1}) - \nabla_\theta \mathcal{J}(\theta^t) \right\|^2 + \frac{L}{\eta} \left\| \theta^{t+1} - \theta^t \right\|^2$$

$$+ \frac{2}{\eta} \left( \mathcal{F}(\theta^{t+1}) - \mathcal{F}(\theta^t) \right) + \frac{2}{\eta} \left\| \nabla_\theta \mathcal{J}(\theta^{t+1}) - \nabla_\theta \mathcal{J}(\theta^t) \right\| \left\| \theta^{t+1} - \theta^t \right\|$$

$$\leq 2 \left\| \nabla_\theta \mathcal{J}(\theta^t) - g^t \right\|^2 + \left( 2L^2 + \frac{3L}{\eta} \right) \left\| \theta^{t+1} - \theta^t \right\|^2 + \frac{2}{\eta} \left( \mathcal{F}(\theta^{t+1}) - \mathcal{F}(\theta^t) \right),$$

where the second inequality is due to the Cauchy-Schwarz inequality and fact that

$$\left\| \nabla_\theta \mathcal{J}(\theta^{t+1}) - g^t \right\|^2 \leq 2 \left\| \nabla_\theta \mathcal{J}(\theta^t) - g^t \right\|^2 + 2 \left\| \nabla_\theta \mathcal{J}(\theta^{t+1}) - \nabla_\theta \mathcal{J}(\theta^t) \right\|^2,$$

and the third inequality is implied by Lemma 4.1.

Summing the above inequality across $t = 0, 1 \ldots, T - 1$, we get

$$\sum_{t=0}^{T-1} \left\| \nabla_\theta \mathcal{J}(\theta^{t+1}) - g^t + \frac{1}{\eta} \left( \theta^{t+1} - \theta^t \right) \right\|^2$$

$$\leq 2 \sum_{t=0}^{T-1} \left\| \nabla_\theta \mathcal{J}(\theta^t) - g^t \right\|^2 + \left( 2L^2 + \frac{3L}{\eta} \right) \sum_{t=0}^{T-1} \left\| \theta^{t+1} - \theta^t \right\|^2 + \frac{2}{\eta} \left( \mathcal{F}(\theta^T) - \mathcal{F}(\theta^0) \right)$$

$$\leq 2 \sum_{t=0}^{T-1} \left\| \nabla_\theta \mathcal{J}(\theta^t) - g^t \right\|^2 + \frac{2}{\eta^2} \sum_{t=0}^{T-1} \left\| \theta^{t+1} - \theta^t \right\|^2 + \frac{2\Delta}{\eta}, \tag{11}$$

where the last inequality is obtained from the fact that $L < \frac{1}{2\eta}$ as a consequence of the choice of the learning rate.

Consequently, we have that

$$\mathbb{E}_T \left[ \operatorname{dist} \left( 0, -\nabla_\theta \mathcal{J}(\hat{\theta}^T) + \partial \mathcal{G}(\hat{\theta}^T) \right)^2 \right]$$

$$= \frac{1}{T} \sum_{t=0}^{T-1} \mathbb{E}_T \left[ \operatorname{dist} \left( 0, -\nabla_\theta \mathcal{J}(\theta^{t+1}) + \partial \mathcal{G}(\theta^{t+1}) \right)^2 \right]$$

$$\leq \frac{1}{T} \sum_{t=0}^{T-1} \mathbb{E}_T \left[ \left\| \nabla_\theta \mathcal{J}(\theta^{t+1}) - g^t + \frac{1}{\eta} \left( \theta^{t+1} - \theta^t \right) \right\|^2 \right]$$

$$\leq \frac{2}{T} \sum_{t=0}^{T-1} \mathbb{E}_T \left[ \left\| \nabla_\theta \mathcal{J}(\theta^t) - g^t \right\|^2 \right] + \frac{2}{\eta^2 T} \sum_{t=0}^{T-1} \mathbb{E}_T \left[ \left\| \theta^{t+1} - \theta^t \right\|^2 \right] + \frac{2\Delta}{\eta T}$$

$$\leq \frac{4}{\eta T (1 - 2\eta L)} \left( \Delta + \frac{1}{2L} \sum_{t=0}^{T-1} \mathbb{E}_T \left[ \left\| g^t - \nabla_\theta \mathcal{J}(\theta^t) \right\|^2 \right] \right) + \frac{2}{T} \sum_{t=0}^{T-1} \mathbb{E}_T \left[ \left\| \nabla_\theta \mathcal{J}(\theta^t) - g^t \right\|^2 \right] + \frac{2\Delta}{\eta T}$$

$$= \left( 2 + \frac{2}{\eta L (1 - 2\eta L)} \right) \frac{1}{T} \sum_{t=0}^{T-1} \mathbb{E}_T \left[ \left\| \nabla_\theta \mathcal{J}(\theta^t) - g^t \right\|^2 \right] + \frac{\Delta}{T} \left( \frac{2}{\eta} + \frac{4}{\eta (1 - 2\eta L)} \right),$$

where the first inequality is because of (7), the second inequality is due to (11) and the third inequality is derived from (9). Thus, the proof is completed. $\qquad\square$

### A.3 Proof of Lemma 4.4

*Proof of Lemma 4.4.* We first estimate $\mathbb{E}_{x \sim \pi_\theta} \left[ \left\| \mathcal{R}_\theta(x) \nabla_\theta \log \pi_\theta(x) + \nabla_\theta \mathcal{R}_\theta(x) \right\|^2 \right]$ as follows

$$\mathbb{E}_{x \sim \pi_\theta} \left[ \left\| \mathcal{R}_\theta(x) \nabla_\theta \log \pi_\theta(x) + \nabla_\theta \mathcal{R}_\theta(x) \right\|^2 \right] \leq 2\mathbb{E}_{x \sim \pi_\theta} \left[ \left\| \mathcal{R}_\theta(x) \nabla_\theta \log \pi_\theta(x) \right\|^2 \right] + 2\mathbb{E}_{x \sim \pi_\theta} \left[ \left\| \nabla_\theta \mathcal{R}_\theta(x) \right\|^2 \right]$$
$$\leq 2U^2 C_g^2 + 2\widetilde{C}_g^2.$$

Then, by the fact that $\mathbb{E} \left[ (X - \mathbb{E}[X])^2 \right] \leq \mathbb{E} \left[ X^2 \right]$ for all random variable $X$, we have

$$\mathbb{E}_{x \sim \pi_\theta} \left[ \left\| \mathcal{R}_\theta(x) \nabla_\theta \log \pi_\theta(x) + \nabla_\theta \mathcal{R}_\theta(x) - \nabla_\theta \mathcal{J}(\theta) \right\|^2 \right] \leq \mathbb{E}_{x \sim \pi_\theta} \left[ \left\| \mathcal{R}_\theta(x) \nabla_\theta \log \pi_\theta(x) + \nabla_\theta \mathcal{R}_\theta(x) \right\|^2 \right]$$
$$\leq 2U^2 C_g^2 + 2\widetilde{C}_g^2,$$

which completes the proof. $\qquad\square$

### A.4 Proof of Theorem 4.5

*Proof of Theorem 4.5.* From Theorem 4.3, in order to ensure that $\hat{\theta}^T$ is a $\epsilon$-stationary point, we can require

$$\left( 2 + \frac{2}{\eta L (1 - 2\eta L)} \right) \mathbb{E}_T \left[ \left\| g^t - \nabla_\theta \mathcal{J}(\theta^t) \right\|^2 \right] \leq \frac{1}{2} \epsilon^2, \quad \forall \, t = 0, \dots, T - 1, \tag{12}$$

$$\frac{\Delta}{T} \left( \frac{2}{\eta} + \frac{4}{\eta (1 - 2\eta L)} \right) \leq \frac{1}{2} \epsilon^2. \tag{13}$$

It is easy to verify that $g^t$ is an unbiased estimator of $\nabla_\theta \mathcal{J}(\theta^t)$. Then, Lemma 4.4 implies that

$$\mathbb{E}_T \left[ \left\| g^t - \nabla_\theta \mathcal{J}(\theta^t) \right\|^2 \right] \leq \frac{\sigma^2}{N}.$$

As a consequence, if one chooses $N = \left\lceil \frac{\sigma^2}{\epsilon^2} \left( 4 + \frac{4}{\eta L (1 - 2\eta L)} \right) \right\rceil$, then (12) holds.

On the other hand, (13) holds if one sets $T = \left\lceil \frac{\Delta}{\epsilon^2} \left( \frac{4}{\eta} + \frac{8}{\eta (1 - 2\eta L)} \right) \right\rceil$. Moreover, we see that the sample complexity can be computed as $TN = O(\epsilon^{-4})$. Therefore, the proof is completed. $\qquad\square$

### A.5 Proof of Lemma 5.3

*Proof of Lemma 5.3.* First, recall that

$$\mathbb{E}_{x \sim \pi_{\theta'}} \left[ \frac{\pi_\theta(x)}{\pi_{\theta'}(x)} \right] = 1.$$

Then, by the definitions of $g$ and $g_w$, we can verify that

$$\mathbb{E}_{x \sim \pi_{\theta'}} \left[ \|g(x, \theta') - g_w(x, \theta, \theta')\|^2 \right]$$

$$\leq 2\mathbb{E}_{x \sim \pi_{\theta'}} \left[ \|g(x, \theta') - g(x, \theta)\|^2 \right] + 2\mathbb{E}_{x \sim \pi_{\theta'}} \left[ \|g(x, \theta) - g_w(x, \theta, \theta')\|^2 \right]$$

$$= 2 \int \|\mathcal{R}_{\theta'}(x) \nabla_\theta \log \pi_{\theta'}(x) - \mathcal{R}_\theta(x) \nabla_\theta \log \pi_\theta(x) + \nabla_\theta \mathcal{R}_{\theta'}(x) - \nabla_\theta \mathcal{R}_\theta(x)\|^2 \pi_{\theta'}(x) \mathrm{d}x$$

$$+ 2 \int \left\| \mathcal{R}_\theta(x) \left( \nabla_\theta \log \pi_\theta(x) - \frac{\pi_\theta(x)}{\pi_{\theta'}(x)} \nabla_\theta \log \pi_\theta(x) \right) + \nabla_\theta \mathcal{R}_\theta(x) - \frac{\pi_\theta(x)}{\pi_{\theta'}(x)} \nabla_\theta \mathcal{R}_\theta(x) \right\|^2 \pi_{\theta'}(x) \mathrm{d}x$$

$$\leq 6 \int \|\mathcal{R}_{\theta'}(x) \left( \nabla_\theta \log \pi_{\theta'}(x) - \nabla_\theta \log \pi_\theta(x) \right)\|^2 \pi_{\theta'}(x) \mathrm{d}x + 6 \int \| \left( \mathcal{R}_{\theta'}(x) - \mathcal{R}_\theta(x) \right) \nabla_\theta \log \pi_\theta(x)\|^2 \pi_{\theta'}(x) \mathrm{d}x$$

$$+ 6 \int \|\nabla_\theta \mathcal{R}_{\theta'}(x) - \nabla_\theta \mathcal{R}_\theta(x)\|^2 \pi_{\theta'}(x) \mathrm{d}x + 4 \int \left\| \mathcal{R}_\theta(x) \left( 1 - \frac{\pi_\theta(x)}{\pi_{\theta'}(x)} \right) \nabla_\theta \log \pi_\theta(x) \right\|^2 \pi_{\theta'}(x) \mathrm{d}x$$

$$+ 4 \int \left\| \nabla_\theta \mathcal{R}_\theta(x) \left( 1 - \frac{\pi_\theta(x)}{\pi_{\theta'}(x)} \right) \right\|^2 \pi_{\theta'}(x) \mathrm{d}x$$

$$\leq \left( 6U^2 C_h^2 + 6C_g^2 \widetilde{C}_g^2 + 6\widetilde{C}_h^2 \right) \|\theta - \theta'\|^2 + \left( 4U^2 C_g^2 + 4\widetilde{C}_g^2 \right) \mathbb{E}_{x \sim \pi_{\theta'}} \left[ \left( \frac{\pi_\theta(x)}{\pi_{\theta'}(x)} - 1 \right)^2 \right]$$

$$= \left( 6U^2 C_h^2 + 6C_g^2 \widetilde{C}_g^2 + 6\widetilde{C}_h^2 \right) \|\theta - \theta'\|^2 + \left( 4U^2 C_g^2 + 4\widetilde{C}_g^2 \right) \left( \int \frac{(\pi_\theta(x))^2}{\pi_{\theta'}(x)} \mathrm{d}x - 1 \right).$$

We next consider the function $f(\theta) := \int \frac{(\pi_\theta(x))^2}{\pi_{\theta'}(x)} \mathrm{d}x$. Taking the derivative of $f$ with respect to $\theta$, we get

$$\nabla_\theta f(\theta) = \int \frac{2\pi_\theta(x) \nabla_\theta \pi_\theta(x)}{\pi_{\theta'}(x)} \mathrm{d}x.$$

Moreover, since

$$\nabla_\theta^2 \log \pi_\theta(x) = \frac{1}{(\pi_\theta(x))^2} \left( \pi_\theta(x) \nabla_\theta^2 \pi_\theta(x) - \nabla_\theta \pi_\theta(x) \nabla \theta \pi_\theta(x)^\top \right)$$

$$= \frac{1}{\pi_\theta(x)} \nabla_\theta^2 \pi_\theta(x) - \nabla_\theta \log \pi_\theta(x) \nabla_\theta \log \pi_\theta(x)^\top,$$

we see that the Hessian of $f$ with respect to $\theta$ can be computed as

$$\nabla_\theta^2 f(\theta) = \int \frac{2}{\pi_{\theta'}(x)} \left( \nabla_\theta \pi_\theta(x) \nabla_\theta \pi_\theta(x)^\top + \pi_\theta(x) \nabla_\theta^2 \pi_\theta(x) \right) \mathrm{d}x$$

$$= \int \frac{2(\pi_\theta(x))^2}{\pi_{\theta'}(x)} \left( 2\nabla_\theta \log \pi_\theta(x) \nabla_\theta \log \pi_\theta(x)^\top + \nabla_\theta^2 \log \pi_\theta(x) \right) \mathrm{d}x.$$

Notice that $f(\theta') = 1$ and $\nabla_\theta f(\theta') = 0$. Therefore, by the Mean Value Theorem, we get

$$f(\theta) = 1 + \frac{1}{2} \left\langle \nabla_\theta^2 f(\tilde{\theta})(\theta - \theta'), \theta - \theta' \right\rangle,$$

where $\tilde{\theta}$ is a point between $\theta$ and $\theta'$. Now, from the expression of the Hessian matrix, we see that for any $\theta \in \mathbb{R}^n$,

$$\left\| \nabla_\theta^2 f(\theta) \right\|_2 \leq \int \frac{2(\pi_\theta(x))^2}{\pi_{\theta'}(x)} \left\| 2\nabla_\theta \log \pi_\theta(x) \nabla_\theta \log \pi_\theta(x)^\top + \nabla_\theta^2 \log \pi_\theta(x) \right\|_2 \mathrm{d}x$$

$$\leq 2(2C_g^2 + C_h) \int \frac{(\pi_\theta(x))^2}{\pi_{\theta'}(x)} dx$$

$$= 2(2C_g^2 + C_h) \left(1 + \mathbb{E}_{x \sim \pi_{\theta'}} \left[ \left( \frac{\pi_\theta(x)}{\pi_{\theta'}(x)} - 1 \right)^2 \right] \right)$$

$$\leq 2(2C_g^2 + C_h)(C_w^2 + 1).$$

As a consequence, we have

$$\mathbb{E}_{x \sim \pi_{\theta'}} \left[ \|g(x, \theta') - g_w(x, \theta, \theta')\|^2 \right]$$

$$\leq \left( 6U^2 C_h^2 + 6C_g^2 \widetilde{C}_g^2 + 6\widetilde{C}_h^2 \right) \|\theta - \theta'\|^2 + \left( 4U^2 C_g^2 + 4\widetilde{C}_g^2 \right) \left( \int \frac{(\pi_\theta(x))^2}{\pi_{\theta'}(x)} dx - 1 \right)$$

$$\leq \left( 6U^2 C_h^2 + 6C_g^2 \widetilde{C}_g^2 + 6\widetilde{C}_h^2 + \left( 4U^2 C_g^2 + 4\widetilde{C}_g^2 \right) (2C_g^2 + C_h)(C_w^2 + 1) \right) \|\theta - \theta'\|^2,$$

which completes the proof. $\qquad\square$

### A.6 Proof of Lemma 5.4

*Proof of Lemma 5.4.* By the definition of the stochastic gradient estimator given in Algorithm 2, we can see that for $t \geq 0$,

$$\mathbb{E}_{t+1} \left[ \|g^{t+1} - \nabla_\theta \mathcal{J}(\theta^{t+1})\|^2 \right]$$

$$= p \mathbb{E}_{t+1} \left[ \left\| \frac{1}{N_1} \sum_{j=1}^{N_1} g(x^{t+1,j}, \theta^{t+1}) - \nabla_\theta \mathcal{J}(\theta^{t+1}) \right\|^2 \right]$$

$$+ (1-p) \mathbb{E}_{t+1} \left[ \left\| \frac{1}{N_2} \sum_{j=1}^{N_2} \left( g(x^{t+1,j}, \theta^{t+1}) - g_w(x^{t+1,j}, \theta^t, \theta^{t+1}) \right) + g^t - \nabla_\theta \mathcal{J}(\theta^{t+1}) \right\|^2 \right]$$

$$= p \mathbb{E}_{t+1} \left[ \left\| \frac{1}{N_1} \sum_{j=1}^{N_1} g(x^{t+1,j}, \theta^{t+1}) - \nabla_\theta \mathcal{J}(\theta^{t+1}) \right\|^2 \right]$$

$$+ (1-p) \mathbb{E}_{t+1} \left[ \left\| \frac{1}{N_2} \sum_{j=1}^{N_2} \left( g(x^{t+1,j}, \theta^{t+1}) - g_w(x^{t+1,j}, \theta^t, \theta^{t+1}) \right) + \nabla_\theta \mathcal{J}(\theta^t) - \nabla_\theta \mathcal{J}(\theta^{t+1}) + g^t - \nabla_\theta \mathcal{J}(\theta^t) \right\|^2 \right]$$

$$\leq p \mathbb{E}_{t+1} \left[ \left\| \frac{1}{N_1} \sum_{j=1}^{N_1} g(x^{t+1,j}, \theta^{t+1}) - \nabla_\theta \mathcal{J}(\theta^{t+1}) \right\|^2 \right]$$

$$+ (1-p) \mathbb{E}_{t+1} \left[ \left\| \frac{1}{N_2} \sum_{j=1}^{N_2} \left( g(x^{t+1,j}, \theta^{t+1}) - g_w(x^{t+1,j}, \theta^t, \theta^{t+1}) \right) + g^t - \nabla_\theta \mathcal{J}(\theta^t) \right\|^2 \right]$$

$$\leq \frac{p\sigma^2}{N_1} + (1-p) \mathbb{E}_{t+1} \left[ \|g^t - \nabla_\theta \mathcal{J}(\theta^t)\|^2 \right] + (1-p) \frac{1}{N_2^2} \sum_{j=1}^{N_2} \mathbb{E}_{t+1} \left[ \|(g(x^{t+1,j}, \theta^{t+1}) - g_w(x^{t+1,j}, \theta^t, \theta^{t+1}))\|^2 \right]$$

$$\leq \frac{p\sigma^2}{N_1} + (1-p) \mathbb{E}_{t+1} \left[ \|g^t - \nabla_\theta \mathcal{J}(\theta^t)\|^2 \right] + \frac{(1-p)C}{N_2} \|\theta^{t+1} - \theta^t\|^2,$$

where in the first inequality, we use the facts that $\mathbb{E}\left[ (X - \mathbb{E}[X])^2 \right] \leq \mathbb{E}\left[ X^2 \right]$ for all random variable $X$ and $g^t$ is unbiased estimator for $\nabla_\theta \mathcal{J}(\theta^t)$ for all $t \geq 0$, in the second inequality, we rely on the fact that

$\{x^{t+1,j}\}$ is independent, and the last inequality is due to Lemma 5.3. By summing the above relation across $t = 0, \ldots, T - 2$, we see that

$$\sum_{t=1}^{T-1} \mathbb{E}_T \left[ \left\| g^t - \nabla_\theta \mathcal{J}(\theta^t) \right\|^2 \right]$$

$$\leq \frac{p\sigma^2(T-1)}{N_1} + (1-p) \sum_{t=0}^{T-2} \mathbb{E}_{t+1} \left[ \left\| g^t - \nabla_\theta \mathcal{J}(\theta^t) \right\|^2 \right] + \frac{(1-p)C}{N_2} \sum_{t=0}^{T-2} \mathbb{E}_T \left[ \left\| \theta^{t+1} - \theta^t \right\|^2 \right],$$

which implies that

$$\sum_{t=0}^{T-1} \mathbb{E}_T \left[ \left\| g^t - \nabla_\theta \mathcal{J}(\theta^t) \right\|^2 \right] \leq \frac{p\sigma^2 T + \sigma^2}{pN_1} + \frac{(1-p)C}{pN_2} \sum_{t=0}^{T-1} \mathbb{E}_T \left[ \left\| \theta^{t+1} - \theta^t \right\|^2 \right]. \tag{14}$$

Recall from (9) that

$$\sum_{t=0}^{T-1} \left\| \theta^{t+1} - \theta^t \right\|^2 \leq \frac{2\eta\Delta}{1 - 2\eta L} + \frac{\eta}{L(1 - 2\eta L)} \sum_{t=0}^{T-1} \left\| g^t - \nabla_\theta \mathcal{J}(\theta^t) \right\|^2,$$

which together with (14) implies that

$$\left( 1 - \frac{(1-p)C\eta}{pN_2 L(1 - 2\eta L)} \right) \sum_{t=0}^{T-1} \mathbb{E}_T \left[ \left\| g^t - \nabla_\theta \mathcal{J}(\theta^t) \right\|^2 \right] \leq \frac{p\sigma^2 T + \sigma^2}{pN_1} + \frac{2\eta(1-p)C\Delta}{pN_2(1 - 2\eta L)}.$$

Thus, the proof is completed. $\qquad\square$

### A.7 Proof Theorem 5.5

*Proof Theorem 5.5.* Since $p = \frac{N_2}{N_1+N_2} \in (0,1)$ and

$$\eta \leq \frac{pN_2 L}{2(1-p)C + 2pN_2 L^2} = \frac{N_2^2 L}{2N_1 C + 2N_2^2 L^2},$$

we can readily check that

$$\eta \in \left( 0, \frac{1}{2L} \right), \quad 1 - \frac{(1-p)C\eta}{N_2 L(1 - 2\eta L)} \geq \frac{1}{2}. \tag{15}$$

Then, we can see that

$$\mathbb{E}_T \left[ \mathrm{dist} \left( 0, -\nabla_\theta \mathcal{J}(\hat{\theta}^T) + \partial \mathcal{G}(\hat{\theta}^T) \right)^2 \right]$$

$$\leq \left( 2 + \frac{2}{\eta L(1 - 2\eta L)} \right) \frac{1}{T} \sum_{t=0}^{T-1} \mathbb{E}_T \left[ \left\| g^t - \nabla_\theta \mathcal{J}(\theta^t) \right\|^2 \right] + \frac{1}{T} \left( \frac{2\Delta}{\eta} + \frac{4\Delta}{\eta(1 - 2\eta L)} \right)$$

$$\leq \frac{1}{T} \left( 2 + \frac{2}{\eta L(1 - 2\eta L)} \right) \left( 1 - \frac{(1-p)C\eta}{pN_2 L(1 - 2\eta L)} \right)^{-1} \left( \frac{p\sigma^2 T + \sigma^2}{pN_1} + \frac{2\eta(1-p)C\Delta}{pN_2(1 - 2\eta L)} \right)$$

$$\quad + \frac{1}{T} \left( \frac{2\Delta}{\eta} + \frac{4\Delta}{\eta(1 - 2\eta L)} \right)$$

$$\leq \frac{4}{T} \left( 1 + \frac{1}{\eta L(1 - 2\eta L)} \right) \left( \frac{T\sigma^2}{N_1} + \frac{(N_1 + N_2)\sigma^2}{N_1 N_2} + \frac{2\eta N_1 C\Delta}{N_2^2(1 - 2\eta L)} \right) + \frac{2\Delta}{T} \left( \frac{1}{\eta} + \frac{2}{\eta(1 - 2\eta L)} \right)$$

where $\Delta := \mathcal{F}^* - \mathcal{F}(\theta^0) > 0$ is a constant, the first inequality is due to Theorem 4.3, the second inequality is derived from Lemma 5.4, and the third inequality is implied by (15).

Then, in order to have $\mathbb{E}_T\left[\text{dist}\left(0, -\nabla_\theta \mathcal{J}(\hat{\theta}^T) + \partial \mathcal{G}(\hat{\theta}^T)\right)^2\right] \leq \epsilon^2$ for a given tolerance $\epsilon > 0$, we can simply set $N_2 = \sqrt{N_1}$,

$$\eta \leq \frac{N_2^2 L}{2N_1 C + 2N_2^2 L^2} = \frac{L}{2C + 2L^2},$$

and require that

$$4\left(1 + \frac{1}{\eta L(1 - 2\eta L)}\right)\frac{\sigma^2}{N_1} \leq \frac{\epsilon^2}{3},$$

$$\frac{4}{T}\left(1 + \frac{1}{\eta L(1 - 2\eta L)}\right)\frac{(N_1 + N_2)\sigma^2}{N_1 N_2} \leq \frac{\epsilon^2}{3},$$

$$\frac{2\Delta}{T}\left[\left(1 + \frac{1}{\eta L(1 - 2\eta L)}\right)\frac{4\eta N_1 C}{N_2^2(1 - 2\eta L)} + \frac{1}{\eta} + \frac{2}{\eta(1 - 2\eta L)}\right] \leq \frac{\epsilon^2}{3}.$$

Therefore, it suffices to set $N_1 = O(\epsilon^{-2})$, $N_2 = \sqrt{N_1} = O(\epsilon^{-1})$ and $T = O(\epsilon^{-2})$. (We ignore deriving the concrete expressions of $T$, $N_1$ and $N_2$, in terms of $\epsilon$ and other constants, but only give the big-O notation here for simplicity.)

Finally, we can verify that the sample complexity can be bounded as

$$N_1 + T\left(pN_1 + (1 - p)N_2\right) = N_1 + T\frac{2N_1 N_2}{N_1 + N_2} \leq N_1 + 2TN_2 = O(\epsilon^{-3}).$$

Therefore, the proof is completed. $\square$

