# OpenReview forum: "On the Stochastic (Variance-Reduced) Proximal Gradient Method for Regularized Expected Reward Optimization"
_TMLR — Accepted by TMLR_

### Review · Reviewer_HEbZ · 2024-06-18

**Summary Of Contributions:**

This paper studies the sample complexity of stochastic gradient methods. In particular, it considers the regularized expected reward optimization problem with general policy parameterization. The paper first analyzes the stochastic proximal gradient method and provides a sample complexity of $O(\epsilon^{-4})$. Then it proposes an efficient stochastic variance-reduced proximal gradient method utilizing an importance sampling-based Probabilistic Gradient Estimator (PAGE), which improves the sample complexity to $O(\epsilon^{-3})$.

**Audience:**

Yes

**Broader Impact Concerns:**

There are no broader impact concerns for this theoretical work.

**Claims And Evidence:**

Yes

**Requested Changes:**

Please address the issues I discussed above.

**Strengths And Weaknesses:**

Strengths

- The setting it considers is more general compared to some other related work, as it considers general convex regularizers, general policy parameterizations, and the non-oblivious setting.

- The sample complexity of $O(\epsilon^{-3})$ analyzed in this paper matches the sample complexity in existing competitive works.

- The proof is well-written and easy to follow.

Major Weaknesses

- The results derived via Algorithm 2 rely on a strong assumption on the importance weight (Assumption 5.1). However, there are papers that achieve the same sample complexity without this assumption (e.g., Fatkhullin et al. 2023, Shen et al. 2019, Yuan et al. 2022). Moreover, although the setting considered in this paper is general, it would be better to compare and discuss the techniques, assumptions, and results in this paper under specific instantiated settings with other works.

- The novelty of the proof techniques is limited, as they are mostly adapted from classical results. For example, the proofs of Lemma 4.1 and Lemma 4.4 are based on the boundedness assumptions; the proof of Theorem 4.4 is based on a standard analysis of the stochastic proximal gradient method; and the proof of Lemma 5.3 is based on the proof of Lemma 4 in Li et al. 2021b. However, I am aware that the TMLR community does not consider novelty a necessary criterion for acceptance.

Minor Weaknesses

- In addition to analyzing global convergence with additional mild assumptions as mentioned in Conclusions, I think it would also be beneficial to analyze the sample complexity for convergence to second-order stationary points, which avoids getting stuck at saddle points.

- The choices of notations $\pi_{\theta}$ (probability density of a single trajectory) and $x$ (a trajectory) are a bit misleading.

- It would be better to present a derivation of $\nabla_{\theta} \mathcal{J}(\theta)$ at the bottom of page 4.

- It would be better if numerical experiments were provided.

- There are assumptions on the Lipschitzness and smoothness of both the reward function(Assumption 3.1) and the score function (Assumption 3.2), whereas most other works with general policy parameterization have at most one of these assumptions for one Theorem.

---

> ### Author Response · Authors · 2024-06-24
> **Authors’ Response to Reviewer HEbZ**
>
> We thank the reviewer HEbZ for valuable comments which have helped to improve the quality of the paper. Below are our detailed responses to the questions. The corresponding modifications are marked in blue color in the revised manuscript, which will be updated after we have addressed the comments from all other reviewers.
>
> # Major Issues
>
> 1. Thanks for the valuable suggestions. Addressing the relaxation of this assumption through the development of a more sophisticated algorithmic framework is beyond the scope of this paper. Following your comments, we have tried to provide more details regarding the Assumption 5.1. In particular, we have overviewed some recent progress, including (Shen et al. 2019, Zhang et al. 2021a, Yuan et al. 2022, Salehkaleybar et al. 2022, and Fatkhullin et al. 2023), on relaxing this stringent condition for MDPs via proposing advanced algorithmic frameworks which build upon the (proximal) policy gradient method. We think these works provide promising future research directions on incorporating these advanced techniques with ours, leading to more valuable results for the general model. We would also like to clarify that we have made extensive efforts to provide detailed comparisons and discussions regarding most of the assumptions, proof techniques, and theoretical results within the specific MDP settings.
>
> 2. Thank you for your insightful comments regarding the novelty of our proof techniques. We acknowledge that our analysis draw upon classical results in the literature. We have made this clarification in the revised manuscript. However, we believe that our work provides a contribution through the application of the stochastic projected gradient methods to a new class of challenging problems, leading to new insights. Moreover, the adaptation of the variance reduction technique via PAGE to the context of regularized nonconvex optimization under the nonoblivious setting is novel, to the best of our knowledge. These adaptations are crucial for addressing the general model, and they enable us to derive the same state-of-the-art convergence properties as in the existing literature under similar conditions. We hope this clarifies how our work builds upon and extends classical approaches in meaningful ways.
>
> # Minor Issues
>
> 1. We agree with the reviewer that establishing global convergence properties without additional assumptions can be challenging, analyzing the sample complexity for achieving convergence to second-order stationary points—thereby avoiding saddle points—may be more realistic and feasible. We have added such a possibility for future research in Conclusions.
>
> 2. Thank you for the careful reading. We have changed the notation used in Example 3.3 to avoid the potential confusion. In particular, a trajectory is denoted by $\tau$ and the probability of a single trajectory is denoted as $\rho_\theta(\tau)$, parameterized by $\theta$.
>
> 3. We have included a derivation of the gradient in the revised text.
>
> 4. Thank you for your suggestion regarding the numerical experiments. We appreciate the importance of empirical validation; however, the primary goal of this work is to provide a rigorous theoretical framework for the (variance-reduced) stochastic proximal gradient methods for solving the interested regularized reward optimization. Incorporating numerical experiments would definitely extend the scope of the paper significantly and could potentially dilute the theoretical contributions we aim to highlight. We think that our theoretical results provide a solid foundation for future numerical studies, and we hope the current presentation is acceptable. Interested readers are referred to (Drusvyatskiy and Xiao 2023) for some numerical studies on the cases when $\mathcal{R}_{\theta}(x)$ is (strongly) convex in both $\theta$ and $x$.
>
> 5. We clarify that most existing works focus on the case when the inner reward function $\mathcal{R}_{\theta}(x)$ does not depend on the parameter $\theta$. For instance, for an MDP, the reward function only depends on states and actions. Hence, the Lipschitzness and smoothness assumptions on the inner reward function are no explicitly needed, and only the boundedness assumption is required. In order to deal with the general model, we think that Assumption 3.1 is also necessary, to the best of our knowledge. We have added some related comments on this assumption in our revised version.

---

> > ### Comment · Reviewer_HEbZ · 2024-07-17
> > **Reviewer HEbZ's Response**
> >
> > Thank you for your responses. I have reviewed the authors' responses, the feedback from all the reviewers, and the revised submission with the highlighted changes. I believe that the authors have addressed most of the issues I raised satisfactorily. Overall, I recommend a borderline accept.

---

### Review · Reviewer_fETn · 2024-06-26

**Summary Of Contributions:**

This paper considers the regularized expected reward optimization problem that subsumes many RL settings (MDPs, bandits, etc). They analyze the standard stochastic proximal gradient approach to get $\epsilon^{-4}$ sample complexity bound. They make further assumptions to show an improved sample complexity bound $\epsilon^{-3}$ for the stochastic variance-reduced proximal gradient approach.

**Audience:**

Yes

**Claims And Evidence:**

Yes

**Requested Changes:**

na

**Strengths And Weaknesses:**

S:

- The paper is well-written and includes relevant related works after all assumptions and results.

W:

As this paper mentions, stochastic variance-reduced gradient approaches exists in optimization by Li et al. (2021b) and MDPs by Gargiani et al. (2022). The results in Section 5 mostly follow the analyses by Gargiani et al. (2022). The additional challenges in this work are the analyses adapted with the parameterized reward functions ($R_\theta$) and regularization function $\mathcal{G}_\theta$. These challenges are mostly overcome by the uniform boundedness assumptions on rewards (and its twice-differentiable functions) and log-policies. So, I'd be more curious about practical implications of this result. That is, showcasing failure of PAGE-PG (Gargiani et al. 2022) in certain toy-problems where adapting to parameterized reward functions and the regularized optimization becomes a necessity.

---

> ### Author Response · Authors · 2024-06-27
> **Authors’ response to Reviewer fETn**
>
> We thank the reviewer fETn for the insightful comments on the comparison with (Gargiani et al. 2022) and the suggestion regarding the numerical experiments. While we appreciate the importance of numerical validations to support our theoretical studies, we consider the present paper a theoretical one and refrain from adding further numerical experiments. Instead, we shall provide the following comparisons with (Gargiani et al. 2022) in the theoretical aspect to support our motivations and contributions.
>
> 1. We argue that studying the regularized optimization is of significant importance, mainly due to the facts that regularization can promote desirable structures in the solutions and can express a certain constraint on the parameter $\theta$. For example, when the direct parameterization of $\pi_\theta$ is adapted in MDPs, then $\pi_\theta$ must subject to certain probabilistic simplex constraints. In this case, the results developed in (Gargiani et al. 2022) are not applicable. One feasible approach to address this issue is to use an indicator function as the regularization term. In this sense, our results generalize existing results on MDPs, leading to more capabilities of the (variance-reduced) policy-gradient-type algorithms. As a conclusion, from the theoretical and modeling aspects, solving the regularized model is in fact natural and necessary.
>
> 2. We acknowledge that our analysis draw upon existing results (e.g., Li et al. 2021b and Gargiani et al. 2022) in the literature. However, we believe that our work provides a contribution through the application of the stochastic proximal gradient methods to a more general model (i.e., regularized nonconvex optimization under the nonoblivious setting), leading to several new insights. Furthermore, the application of PAGE in the context of this general model is novel, to the best of our knowledge. These adaptations are crucial for deriving the same state-of-the-art convergence properties (developed for MDPs) as in the existing literature under similar conditions. We hope this clarifies how our work builds upon and extends classical approaches in meaningful ways.
>
> 3. Our paper studies a more general model than the so-called performative prediction, in which we do not assume $\mathcal{R}$ to be strongly convex. Interested readers are referred to (Drusvyatskiy and Xiao 2023) for some numerical studies on the cases when $\mathcal{R}_{\theta}(x)$ is (strongly) convex in both $\theta$ and $x$.

---

### Review · Reviewer_LAnA · 2024-07-02

**Summary Of Contributions:**

In this paper, the authors introduce formally a general problem setup for reward optimization. Authors introduce the problem setup, establish the smoothness for part of the objective under some assumptions, propose to solve the problem using existing proximal gradient descent methods and provide a convergence guarantee for the algorithm, with a sample complexity of $O(\epsilon^{-4})$ to attain a $\epsilon-$stationary point. The paper also provide a  stochastic variance reduced version by applying existing variance reduction method PAGE, which has an improved sample complexity of $O(\epsilon^{-3})$.

**Audience:**

Yes

**Claims And Evidence:**

Yes

**Requested Changes:**

As mentioned in the Weaknesses section, it seems important to better motivate the general problem setup introduced in the paper, with motivating practical applications and/or some illustrative example that demonstrate the general problem setup introduced in this paper, which seems to be a critical requirement for the paper. Please refer to Minor Weaknesses for minor changes needed in the paper.

**Strengths And Weaknesses:**

Strengths

* The problem setup introduced in this paper seems to be novel and not discussed in prior literature
* The presentation of the problem, assumptions on the problem setup, method to solve the problem and related theoretical results are presented clearly and the paper is fairly easy to follow

Weaknesses

* Since the problem setup is novel (which seems to be one of the main contributions of the paper) and not discussed in prior literature, the problem setup should be well motivated with several application setups and/or examples, which seems to be lacking.
* Related to previous point, it is unclear when one would need to optimize the policy and the reward function which shares the same parameter. In general, it seems like this kind of setup might lead to learning a meaningless reward function which the policy learn to maximize.
* The assumption of the boundedness of the norm of the gradient and the Hessian of the reward function seems to need more discussion and justification, since this kind of assumption is new and seems to be related to the novel problem setup introduced in this paper.
The example provided in the paper as a illustration of the setup is a special case of the proposed problem setup, which is not illustrative of the novel setup and widely discussed in reinforcement learning (RL) literature.
* There are some strong assumptions used for analyzing the variance reduced method, and the paper lacks practical application of the proposed method which might bolster the validity of the novel setup introduced in the paper. The authors acknowledge these limitations in the paper.

Minor Weaknesses:

* $\theta^* =$ might be incorrect in Equation (2) since the solution to problem in Equation (2) might not be unique.
* The subsequent simplifications of the original problem used in Example 3.3 use $\min$ operator, which is inconsistent with the original problem formulation.
* The definition of sub gradient given after Equation (3) seems to be the definition of a subdifferential.
* The function $\text{dist}(\cdot, \cdot)$ used in Definition 3.4 is not properly introduced/defined.

---

> ### Author Response · Authors · 2024-07-04
> **Authors’ response to Reviewer LAnA**
>
> We thank the reviewer LAnA for the valuable comments which have helped to improve the quality of this paper. Below is our detailed response. The modifications (marked in blue) and some new references can be found in the revised text.
>
> # Major Issues
>
>  ## Regarding the significance of the interested model
>
> Thanks for the suggestions regarding providing more illustrative applications to support the general model in the paper. Below are our clarifications and modifications in addressing these concerns.
>
> - When applied to MDPs with convex regularization, the variance reduced stochastic proximal policy-gradient method with PAGE turns out to be a novel approach, to the best of our knowledge. This could be one of the contributions of our work.
>
> - Though we used the convention in RL, the interested model is in fact a nonconvex composite optimization problem in the non-oblivious setting, which is closely related to the subjects of performative prediction and stochastic optimization with decision-dependent distributions that attract increasing attentions recently; see e.g., (Mendler-Dünner et al.,2020; Perdomo et al. 2020; Drusvyatskiy & Xiao, 2023) and references therein. From these works, we recognize that some potential applications of the model may include:
>     - Concept drift (Gama et al., 2014): the problem of learning when the target distribution over instances drifts over time. It refers to the phenomenon where the statistical properties of the target variable that a model is trying to predict change over time. This can lead to a degradation in the performance of the model because the patterns it learned during training may no longer be valid. Concept drift is particularly relevant in dynamic environments where the underlying data generating processes are not static.
>
>     - Strategic classification (Tsirtsis et al., 2024): a subject of machine learning and game theory that involves scenarios where individuals or entities may alter their behavior or characteristics in response to the classification system to achieve a more favorable outcome. This is particularly relevant in applications where decisions are made based on predictive models, such as credit scoring, hiring processes, or fraud detection. For example, (Milli et al.,2019) suggested that strategy-robustness must be weighed against considerations of social welfare and fairness.
>
>     - Casual inference (Yao et al., 2021): a field within statistics and data science that focuses on determining whether a cause-and-effect relationship exists between variables. Unlike traditional statistical analysis, which often focuses on correlation, causal inference aims to establish causality, answering questions about what happens to one variable when another variable is manipulated.
>
> We have included such potential applications of the model in the introduction. We hope that the above discussion provides strong evidence for the significance of the interested optimization model.
>
> ## Regarding the boundedness conditions
>
> We argue that the boundedness conditions date back to (Sutton et al., 1999) which sets up the theoretical foundation of modern policy gradient type methods that are prevalent in the RL and ML literature. We clarify that the boundedness condition of the gradient is used for ensuring that the gradient of J is well-defined for any $\pi_{\theta}$. And the boundedness condition of the Hessian is used to guarantee the Lipschitz continuity of the gradient of $J$. Without the well-definiteness and the L-smoothness of $J$, convergence analysis for the (stochastic) gradient-type methods becomes extremely difficult. In fact, without assuming other stronger conditions, we think that it is still an open question if one can get the same convergence properties for the stochastic gradient methods as shown in this paper, to the best of our knowledge.
>
> ## Regarding the bounded variance of importance weight
>
> We agree with the reviewer that the boundedness condition of the importance weight is quite strong and may not be verified in general. There are several existing works focusing on addressing this issue.  We have overviewed some recent progress, including (Shen et al. 2019, Zhang et al. 2021a, Yuan et al. 2022, Salehkaleybar et al. 2022, and Fatkhullin et al. 2023), on relaxing this stringent condition for MDPs via proposing advanced algorithmic frameworks which build upon the (proximal) policy gradient method. Please refer to the changes after Assumption 5.1. We think these works provide promising future research directions on incorporating these advanced techniques with ours, leading to more valuable results for the general model.
>
> # Minor Issues
>
> - We have changed $=$ to $\in$ to indicate that the optimizer may not be unique.
>
> - We have corrected the mistakes and changed “min” to “max”.
>
> - We agree with the reviewer that the term “subdifferential” is more appropriate. We have changed the name.
>
> - We have added the definition of “dist” in Definition 3.4.

---

> > ### Comment · Reviewer_LAnA · 2024-07-17
> >
> > Thank you for the response. The response and the updated paper address my concerns in a satisfactory manner. In light of this, I recommend to accept the paper.

---

### Decision · Action_Editor_fCCu · 2024-08-15

**Recommendation:** Accept with minor revision

**Comment:**

We have one Accept and two Leaning Accepts among reviewers. I also read the paper before closely reading their reviews, and I agree with them that this is a reasonably good paper: it has some non-trivial contributions, it is generally well-written, and according to reviewers, sufficient evidence is provided for its claims (I have not verified the proofs myself).

A major novelty of this work is that it considers a new class of problems in which both the policy and the reward function are parameterized. To solve this problem, a policy gradient-based approach, specifically the stochastic proximal gradient algorithm, is proposed to solve the regularized objective. The convergence rate to the stationary point is established. Furthermore, the paper suggests the use of a variance-reduction algorithm, PAGE. This leads to an improved convergence rate, under extra assumptions.

It is notable that these types of convergence analysis for policy gradient-based methods have become relatively common in the past few years, but since this is a new problem setup, I would consider it a sufficiently novel and interesting contribution, even if some of the proof techniques have been used in the past.

The authors have revised the paper to address several of the reviewers comments. I believe they have done a reasonable job.

I recommend **acceptance with minor revisions**. Before accepting this paper, I would like to ask the authors to do a few other minor revisions in order to improve the readability of this work:

- Please clarify the use of x a bit further. Specifically, is $x$ the same as $\tau$ in Example 3.3?

- Assumption 5.1 is strong, as Remark 5.2 already explains. What is not clear is how strong it is. It would be very helpful if the authors provide an analysis of the size of constant $C_w$ in that assumption. This can be done for a simple concrete example, if a general characterization is difficult.
For MDPs, this constant seems to depend on the ratio of probability of actions according to each policy along the whole trajectory. I can see that if there is no restriction over $\theta$ and $\theta’$, that is, they can be far from each other, this ratio can be very large, but if they happen to be in the same neighbourhood, they may be bounded, perhaps under some smoothness of the policy as a function of parameter theta (this might already be implied by Assumption 3.2). A discussion and a worked out example should not be very difficult but goes a long way.

- Even though PAGE is not introduced in this work, its use is one of the contributions of this paper. Currently there is not much intuition of why PAGE helps. This makes the paper a bit mysterious to a reader who has not read the original PAGE paper. I’d ask the authors to provide some intuition behind that algorithm, so that this paper would be more self-contained.

**Audience:**

Yes, this paper would be of interest to the reinforcement learning researchers, as well as those interested in stochastic optimization with decision-dependent distributions.

**Claims And Evidence:**

Yes, all reviewers believe that the paper has provided sufficient evidence for its claims.

---

> ### Author Response · Authors · 2024-08-18
> **Response to Action Editor fCCu**
>
> We appreciate the valuable suggestions provided by the action editor, which we think are very helpful for improving the paper. Below are our responses to the questions.
>
> 1. In Example 3.3, $\tau$ and $\rho_\theta$ correspond to $x$ and $\pi_\theta$ in problem (1), respectively. Following your suggestion, we have added a footnote to highlight this fact. Please see our modifications near equation (2).
>
> 2. Indeed, it is important to get some insights on how significant the constant $C_w$ in the Assumption 5.1 is. Following your suggestion, we have provided some detailed discussions on the case when $\pi_\theta = \theta$ is a discrete distribution over a set of finite point $\\{x_k\\}$. One can observe that the variance in Assumption 5.1 is controlled by $\sum_{k=1}^n\theta_k/\theta_k'$ which depends sensitively on the values of $\theta$ and $\theta'$. Thus, in general the constant $C_w$ may not exist or can be extremely large since it may be the case when $\theta_k'$ is small or zero. Fortunately, since we can use the regularization term $G(\theta)$ to impose a bounded constraint on $\theta$, say, $\theta_k \geq \delta > 0$ for all $k$. In this way, one can ensure that $\sum_{k=1}^n\theta_k/\theta_k' \leq 1/\delta$. Please refer to our modifications between Assumption 5.1 and Remark 5.2.
>
> 3. Thank you for your suggestion on making the paper more self-contained. We have included more discussions on PAGE in Section 5. Please see our modifications made in the second paragraph in Section 5.